# Dengue viremia kinetics and effects on platelet count and clinical outcomes: An analysis of 2340 patients from Vietnam

Nguyen Lam Vuong[1,2]*, Nguyen Than Ha Quyen[2], Nguyen Thi Hanh Tien[2], Kien Duong Thi Hue[2], Huynh Thi Le Duyen[2], Phung Khanh Lam[1,2], Dong Thi Hoai Tam[2], Tran Van Ngoc[3], Thomas Jaenisch[4,5], Cameron P Simmons[6,7], Sophie Yacoub[2,6], Bridget A Wills[2,6], Ronald Geskus[2,6]*

[1]University of Medicine and Pharmacy at Ho Chi Minh City, Ho Chi Minh City, Viet Nam; [2]Oxford University Clinical Research Unit, Ho Chi Minh City, Viet Nam; [3]Hospital for Tropical Diseases, Ho Chi Minh City, Viet Nam; [4]Center for Global Health, Colorado School of Public Health, Aurora, United States; [5]Heidelberg Institute of Global Health (HIGH), Heidelberg University Hospital, Heidelberg, Germany; [6]Centre for Tropical Medicine and Global health, Nuffield Department of Clinical Medicine, University of Oxford, Oxford, United Kingdom; [7]World Mosquito Program, Monash University, Clayton, Australia

*For correspondence:
vuongnl@oucru.org (NLV);
rgeskus@oucru.org (RG)

## Abstract

**Background:** Viremia is a critical factor in understanding the pathogenesis of dengue infection, but limited data exist on viremia kinetics. This study aimed to investigate the kinetics of viremia and its effects on subsequent platelet count, severe dengue, and plasma leakage.

**Methods:** We pooled data from three studies conducted in Vietnam between 2000 and 2016, involving 2340 dengue patients with daily viremia measurements and platelet counts after symptom onset. Viremia kinetics were assessed using a random effects model that accounted for left-censored data. The effects of viremia on subsequent platelet count and clinical outcomes were examined using a landmark approach with a random effects model and logistic regression model with generalized estimating equations, respectively. The rate of viremia decline was derived from the model of viremia kinetics. Its effect on the clinical outcomes was assessed by logistic regression models.

**Results:** Viremia levels rapidly decreased following symptom onset, with variations observed depending on the infecting serotype. DENV-1 exhibited the highest mean viremia levels during the first 5–6 days, while DENV-4 demonstrated the shortest clearance time. Higher viremia levels were associated with decreased subsequent platelet counts from day 6 onwards. Elevated viremia levels on each illness day increased the risk of developing severe dengue and plasma leakage. However, the effect size decreased with later illness days. A more rapid decline in viremia is associated with a reduced risk of the clinical outcomes.

**Conclusions:** This study provides comprehensive insights into viremia kinetics and its effect on subsequent platelet count and clinical outcomes in dengue patients. Our findings underscore the importance of measuring viremia levels during the early febrile phase for dengue studies and support the use of viremia kinetics as outcome for phase-2 dengue therapeutic trials.

**Funding:** Wellcome Trust and European Union Seventh Framework Programme.

## eLife assessment

This manuscript by Vuong and colleagues reports on the kinetics of viremia in a large set of individuals from Vietnam. In the large cohort, all 4 dengue serotypes are represented and the authors try to correlate viraemia measured at various days from illness onset with thrombocytopaenia and severe dengue, according to the WHO 2009 classification scheme. These are **fundamental** findings that provide **compelling** evidence of the importance of measuring viremia early in the phase of the disease. These data will help to inform the design of studies of antiviral drugs against dengue.

## Introduction

Dengue is the most common arboviral infection worldwide and is found mainly in tropical and subtropical areas. The disease has been a growing threat for decades, and in 2019 the World Health Organization (WHO) ranked it among the top 10 threats to global health (*World Health Organization, 2009*; *World Health Organization, 2020*). It is estimated that 105 million people are infected by dengue annually, of whom half develop symptoms (*Cattarino et al., 2020*). Although most infections are asymptomatic or self-limiting, the small proportion of patients who develop complications can still overwhelm health systems in dengue-endemic countries during seasonal epidemics. Climate change, increasing global travel, and urbanization all serve to amplify the distribution of *Aedes aegypti*, the main vector responsible for dengue transmission (*Whitehorn and Yacoub, 2019*; *Yacoub et al., 2011*). Although dengue has been known for centuries, specific treatment is still to be found, and the two licensed vaccines are of limited benefit (*Kariyawasam et al., 2023*; *Redoni et al., 2020*).

Exploring the variation in viremia levels could help improve our understanding of disease pathogenesis and potentially facilitate assessment of new vaccines and treatments for dengue. For example, knowing the natural kinetics of dengue viremia may help in selection of patient groups or timing of antiviral use in therapeutic intervention trials. Dengue viremia kinetics has been investigated in several studies (*Ben-Shachar et al., 2016*; *Clapham et al., 2016*; *Clapham et al., 2014*; *Duyen et al., 2011*; *Matangkasombut et al., 2020*; *Nguyet et al., 2013*; *Simmons et al., 2007a*; *Tricou et al., 2011*). The main findings are: (1) viremia rapidly decreases after symptom onset; (2) dengue virus (DENV)-1 infection gives higher viremia level than DENV-2 and DENV-3; and (3) primary infection gives higher viremia than secondary infection in DENV-1. However, these studies had limited sample sizes, particularly for DENV-4 serotype, and did not provide a full account of viremia kinetics. Three studies constructed mechanistic mathematical models and focused on investigating the role of the host immune response in controlling viremia (*Ben-Shachar et al., 2016*; *Clapham et al., 2016*; *Clapham et al., 2014*). Another study assessed viremia in infants aged under 18 months (*Simmons et al., 2007a*); the findings cannot be generalized to older children or adults. Others focused on peak viremia level after symptom onset, time to viral clearance and/or the probability of detecting virus on each illness day (*Duyen et al., 2011*; *Matangkasombut et al., 2020*; *Nguyet et al., 2013*; *Tricou et al., 2011*). In most analyses, viremia levels below the limit of detection were simply set as zero, which may not be appropriate.

Quantifying the effect of viremia level on clinical outcomes is important to understand the mechanisms leading to severe disease. We and others have previously found that higher viremia levels are associated with more severe dengue (*Duyen et al., 2011*; *Endy et al., 2004*; *Morsy et al., 2020*; *Tang et al., 2010*; *Tricou et al., 2011*; *Vaughn et al., 2000*; *Vuong et al., 2021*), but the effect size was not strong (*Vuong et al., 2021*). However, all analyses were based on a single viremia value per individual. Investigating how individual viremia levels at different times after symptom onset affect the risk of severe outcome potentially provides new insights into pathogenic mechanisms. Another important question is whether a more rapid decline in viremia is associated with a lower risk of more severe clinical outcomes. One study has shown a slower rate of viral clearance being associated with more severe outcomes (*Wang et al., 2006*), while others have shown no association (*Fox et al., 2011*; *Vaughn et al., 2000*). This lack of conclusive evidence underscores the need for further investigation. In addition, the relationship between viremia kinetics and platelet count kinetics – an important hematological indicator – has yet to be investigated.

This study aims to improve our understanding of viremia kinetics, how it is associated with patient characteristics and virus serotype, and how it impacts disease severity, using a dataset derived from several thousand individuals with symptomatic dengue. First, we fitted a model for individual viremia kinetics that treats values below the detection limit as censored data, and assessed whether viremia kinetics differed by age, sex, serotype, or immune status. Second, we modeled platelet count kinetics

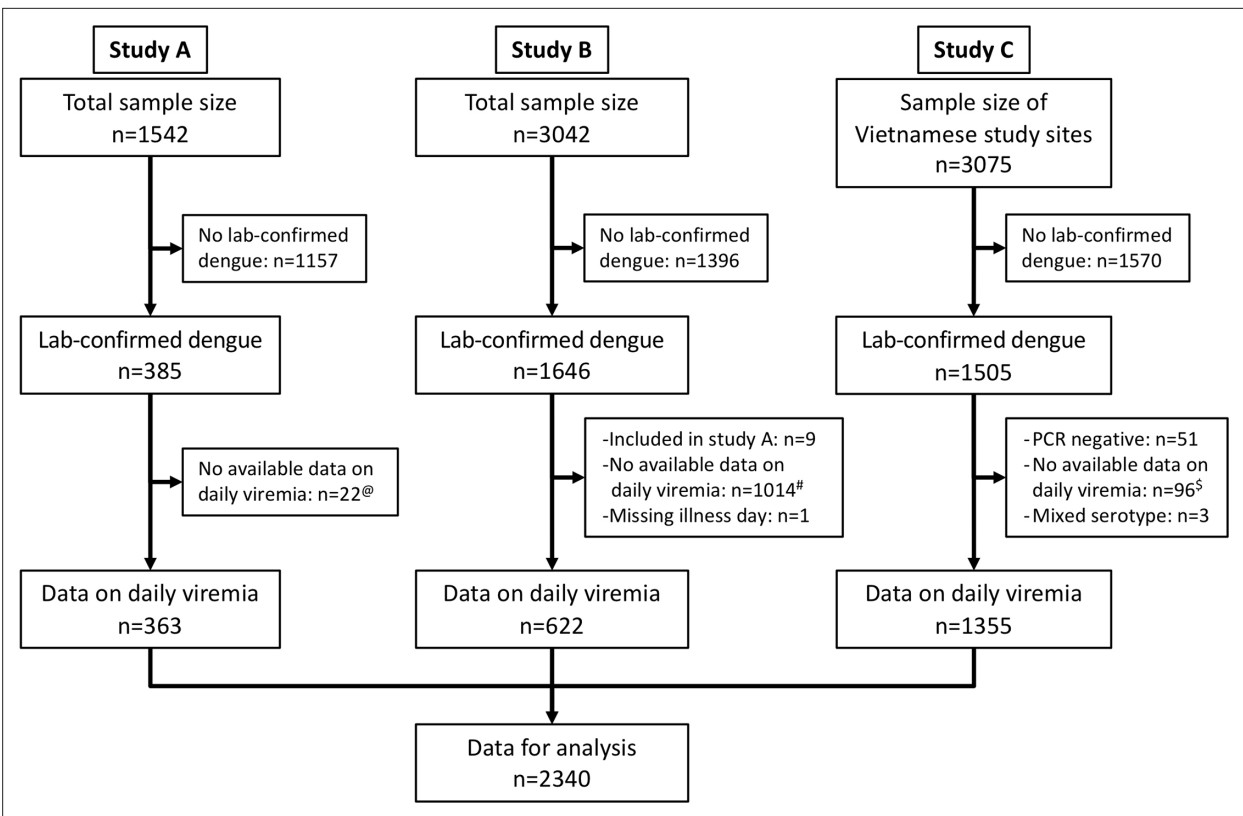

**Figure 1.** Flowchart illustrating the process of patient selection. [@]Among 385 laboratory-confirmed dengue patients in study A, 31 individuals were later included in study B upon hospitalization. Of these, nine had viremia measurements available in both studies and were consequently analysed in study A. The remaining 22 lacked viremia data in study A but had measurements in study B, leading to their inclusion in study B in the analysis. [#]Although study B spanned from 2000 to 2009, daily viremia levels were only measured between 2006 and 2008. [$]In study C, one study site in Vietnam did not have daily viremia measurements. Overall, a random sample of 2340 from 3527 laboratory-confirmed dengue patients had data on viremia kinetics. PCR, polymerase chain reaction.

and explored dependence on viremia level. We investigated the potential existence of a lagged effect of viremia, by analyzing platelet counts measured on and following the day of viremia assessment. Third, we modeled the effect of viremia levels measured at different days after symptom onset and the effect of the rate of viremia decline on development of two clinical outcomes, plasma leakage and severe dengue.

## Methods

### Study population

We used data from three prospective observational studies performed as part of the longstanding collaboration between the Oxford University Clinical Research Unit (OUCRU) and the Hospital for Tropical Diseases (HTD) in Ho Chi Minh City, Vietnam. Protocol synopses for the three studies are presented in Appendix 1. Briefly, studies A and B simultaneously ran from 2000 to 2009 and enrolled children aged between 5 and 15 years. Study A included children with a febrile illness presenting to community outpatient clinics, while study B included children admitted to HTD with a febrile illness suspected to be dengue. Some patients from study A were subsequently enrolled in study B after admission to hospital. Study C ran from 2011 to 2016 and enrolled adults and children presenting with possible dengue within 3 days of fever onset (IDAMS study, NCT01550016) (*Jaenisch et al., 2016*). Clinical evaluation and blood sampling were performed daily for all three studies. All studies were approved by the Oxford Tropical Research Ethics Committee and the Institutional Ethics Committee.

We identified 3527 laboratory-confirmed dengue patients. Among them, 2340 had relevant data on viremia kinetics and were included in this analysis (*Figure 1*). Details of dengue diagnostics are in Appendix 2.

## Plasma viremia measurement, dengue diagnostics, and clinical endpoints (details in Appendix 2)

Plasma viremia levels were measured by reverse transcription polymerase chain reaction (RT-PCR), which was an internally controlled, serotype-specific, real-time, two-step assay in studies A and B (*Simmons et al., 2007b*), and a one-step procedure using a validated assay in study C (*Hue et al., 2011*). There was no formal validation of the detection limit for viremia for the two-step PCR used in studies A and B; we set it at 1000 copies/ml. In the one-step PCR used in study C, the detection limit was 300 copies/ml for DENV-1 and DENV-3, 60 copies/ml for DENV-2, and 600 copies/ml for DENV-4 (*Hue et al., 2011*).

Patients were classified into probable primary (i.e., the first) or probable secondary (i.e., a second or subsequent) infection based on IgG results on paired samples. A probable primary infection was defined by two negative/equivocal IgG results on separate samples taken at least 2 days apart within the first 10 days of symptom onset, with at least one sample during the convalescent phase (days 6–10). A probable secondary infection was defined by at least one positive IgG result during the first 10 days. Cases without time-appropriate IgG results were classified as indeterminate.

For the clinical endpoints we selected severe dengue and plasma leakage. The definitions are based on the WHO 2009 guidelines (*World Health Organization, 2009*) and standard endpoint definitions for dengue trials (*Tomashek et al., 2018*). Severe dengue was defined as severe plasma leakage, severe bleeding, and/or severe organ impairment. Plasma leakage included moderate and severe leakage.

## Statistical analysis (details in Appendix 3)

In *Appendix 3—figure 1* we show a directed acyclic graph (DAG) to display assumptions about the causal relationships between variables over illness day. Illness day is the number of days after symptom onset, where day 1 is the day of symptom onset. Potential confounders that needed to be corrected for were age, sex, DENV serotype, and primary/secondary immune status. In all analyses, we used a log-10 transformation for viremia levels.

We modeled viremia kinetics over time and explored dependence on age, sex, serotype, immune status, and PCR method using a linear mixed-effects regression model, allowing for left-censored values in the outcome variable. In the fixed effect, we allowed for interactions between serotype and immune status, and between illness day and all other covariates. Nonlinear trends by age and illness day were investigated using splines with three and four knots, respectively. We included a random effect with an intercept and splines for illness day with four knots to model the intra-person correlation. Due to the presence of a left-skewed distribution in log-10 viremia (*Appendix 4—figure 1*), a sensitivity analysis was conducted using the 10th-root transformation.

Platelet counts were transformed using a fourth-root transformation since the distribution was right-skewed (*Appendix 4—figure 2*). The effect of viremia on subsequent platelet count was investigated using the landmark approach (*van Houwelingen and Putter, 2011*). For each illness day from 1 to 7, a landmark dataset was created, which included viremia on that day, platelet count from that day to day 10, and the time-fixed variables (age, sex, serotype, immune status, and PCR method). For the supermodel that combined all landmark datasets, we used a linear mixed-effects model including viremia, age, sex, serotype, immune status, PCR method, illness day and landmark. In the fixed effect, we also included four-way interactions between viremia, serotype, PCR method and (1) and immune status, (2) illness day, and (3) landmark day. Splines with three knots were used to allow for nonlinear trends in log-10 viremia, age, illness day, and landmark day. We included a random effect with an intercept by the combination of individuals and landmark, and splines for illness day with three knots.

The effect of viremia on the clinical endpoints was investigated using a logistic regression model using the landmark approach (*van Houwelingen and Putter, 2011*). For each illness day from 1 to 7, a landmark dataset was created, which included viremia on that day, occurrence of the endpoint on or after that day, and the time-fixed variables similar to above. All individuals who had already experienced the relevant endpoint before that day were excluded. We fitted a logistic regression model for

each landmark dataset and then fitted a supermodel that combined all the datasets. For the supermodel we used generalized estimating equations to account for repeated inclusion of individuals at multiple landmarks. We included viremia, age, sex, serotype, immune status, study, and landmark day, and allowed for the four-way interaction between viremia, serotype, immune status, and landmark day, and two-way interactions between viremia and all other covariates. Splines were used to allow for nonlinear trends in log-10 viremia (three knots), age (three knots), and landmark (three knots). Since plasma leakage had missing data, we performed an analysis with imputed data using multiple imputation by chained equations. The imputation was done in our previous study (*Vuong et al., 2021*).

To calculate the rate of viremia decline for each patient, we assumed a linear decline in log-10 viremia over time. We modified our viremia kinetics model by using a linear term for illness day rather than the original nonlinear terms, both in the fixed and in the random effect. For each patient, we then calculated fitted values using this model. The rate of viremia decline was defined as the daily difference in log-10 viremia. We assessed the impact of the decline rate on the clinical endpoints using logistic regression models. We did not use the landmark approach for this analysis since the decline rate was assumed to be constant over time. Covariates included the decline rate, serotype, immune status, and PCR method. Due to limited number of severe dengue cases, its model did not include any interaction terms. The plasma leakage model incorporated interactions between the decline rate and all other covariates.

In the analyses for platelet count and clinical outcomes, undetectable viremia levels were set as the specific detection limits, and a binary dummy variable (yes/no) was created to indicate whether the viremia value was undetectable. Since the individual spline terms are hard to interpret and the large dataset makes most differences statistically significant, results are primarily reported using plots of predicted values. All analyses were done using the statistical software R version 4.1.0 (*R Core Team, 2021*) with 'MCMCglmm' (*Hadfield, 2010*), 'lme4' (*Bates et al., 2015*), 'geepack' (*Halekoh et al., 2006*), and 'rms' (*Harrell, 2021*) packages for the models.

## Results

Baseline characteristics and outcomes are summarized in *Table 1*. Three quarters of the participants were children, and male sex predominated (60%). Most patients (85%) were enrolled on illness day 2 or 3. All four serotypes were included, but DENV-1 was the most prevalent (54%). Most infections (69%) were classified as probable secondary infection, while in 11% immune status could not be determined. There were 353 patients (15%) with plasma leakage and 65 patients (3%) with severe dengue. In *Appendix 4—table 1*, we summarize plasma leakage and severe dengue by serotype and immune status. DENV-2 had the highest proportion of these two outcomes. Patients with probable secondary infection were more likely to experience these outcomes than those with probable primary infection or indeterminate immune status, regardless of the infecting serotype.

### Viremia kinetics and the relationship with clinical characteristics

Individual viremia trajectories are shown in *Figure 2*. Most individuals had a decreasing trend from day 1 or 2 onwards. However, values higher than the baseline value were observed up to day 7 (*Appendix 4—figure 3*). The decreasing trend was consistent across all combinations of serotype and immune status (*Figure 2*).

In *Figure 3*, we present the fitted values based on the model for viremia for the one-step PCR cohort. The mean viremia trajectory differed by serotype. DENV-1 gave the highest viremia levels, with DENV-2 in secondary infection giving similar levels from day 5 onwards. DENV-2, DENV-3, and DENV-4 had similar viremia levels during the first 3 days. However, viremia decreased rapidly in DENV-4, reaching undetectable levels in the shortest time, while DENV-2 showed the slowest decline. These differences between serotypes were more pronounced in case of probable primary infection (*Figure 3A*). In terms of immune status, probable primary infection showed higher viremia levels compared to probable secondary infection in DENV-1 from day 3 onwards. The disparity between probable primary and probable secondary infection was less pronounced in the other serotypes (*Figure 3B*). Mean viremia levels were comparable between females and males (*Figure 3C*), as well as across age (*Figure 3D*). The one-step PCR method resulted in longer viremia compared to the two-step PCR (*Appendix 5—figure 1*). P values for the differences by age, serotype, immune status,

**Table 1.** Baseline characteristics and clinical outcomes.

| | N* | All patients (N = 2340) | Study A (N = 363) | Study B (N = 622) | Study C (N = 1355) |
|---|---|---|---|---|---|
| Age, years | 2340 | 13 (10; 18) | 12 (9; 14) | 12 (10; 13) | 16 (10; 25) |
| Sex male | 2340 | 1403 (60) | 195 (54) | 412 (66) | 796 (59) |
| Illness day at enrolment | 2340 | | | | |
| 1 | | 310 (13) | 59 (16) | 7 (1) | 244 (18) |
| 2 | | 848 (36) | 150 (41) | 155 (25) | 543 (40) |
| 3 | | 1137 (49) | 114 (31) | 455 (73) | 568 (42) |
| 4 | | 45 (2) | 40 (11) | 5 (1) | 0 (0) |
| Serotype | 2340 | | | | |
| DENV-1 | | 1264 (54) | 233 (64) | 410 (66) | 621 (46) |
| DENV-2 | | 373 (16) | 48 (13) | 130 (21) | 195 (14) |
| DENV-3 | | 252 (11) | 80 (22) | 82 (13) | 90 (7) |
| DENV-4 | | 451 (19) | 2 (1) | 0 (0) | 449 (33) |
| Immune status | 2340 | | | | |
| Probable primary | | 474 (20) | 134 (37) | 124 (20) | 216 (16) |
| Probable secondary | | 1619 (69) | 219 (60) | 464 (75) | 936 (69) |
| Indeterminate | | 247 (11) | 10 (3) | 34 (5) | 203 (15) |
| Plasma leakage | 2288 | 353 (15) | 43 (13) | 177 (29) | 133 (10) |
| Missing | | 52 | 39 | 13 | 0 |
| Illness day of plasma leakage | 327 | | | | |
| 3 | | 2 (1) | 0 (0) | 0 (0) | 2 (2) |
| 4 | | 71 (22) | 12 (30) | 35 (23) | 24 (18) |
| 5 | | 107 (33) | 12 (30) | 50 (32) | 45 (34) |
| 6 | | 105 (32) | 12 (30) | 49 (32) | 44 (33) |
| 7 | | 42 (13) | 4 (10) | 20 (13) | 18 (14) |
| Missing | | 26 | 3 | 23 | 0 |
| Severe dengue | 2340 | 65 (3) | 6 (2) | 40 (6) | 19 (1) |
| Illness day of severe dengue | 65 | | | | |
| 3 | | 2 (3) | 0 (0) | 2 (5) | 0 (0) |
| 4 | | 13 (20) | 0 (0) | 12 (30) | 1 (5) |
| 5 | | 24 (37) | 2 (33) | 16 (40) | 6 (32) |
| 6 | | 20 (31) | 4 (67) | 7 (18) | 9 (47) |
| 7 | | 6 (9) | 0 (0) | 3 (8) | 3 (16) |

Summary statistics are number of patients (%) or median (25th; 75th percentiles).

DENV: dengue virus.

*N represents the number of patients with available data (i.e., without missing data).

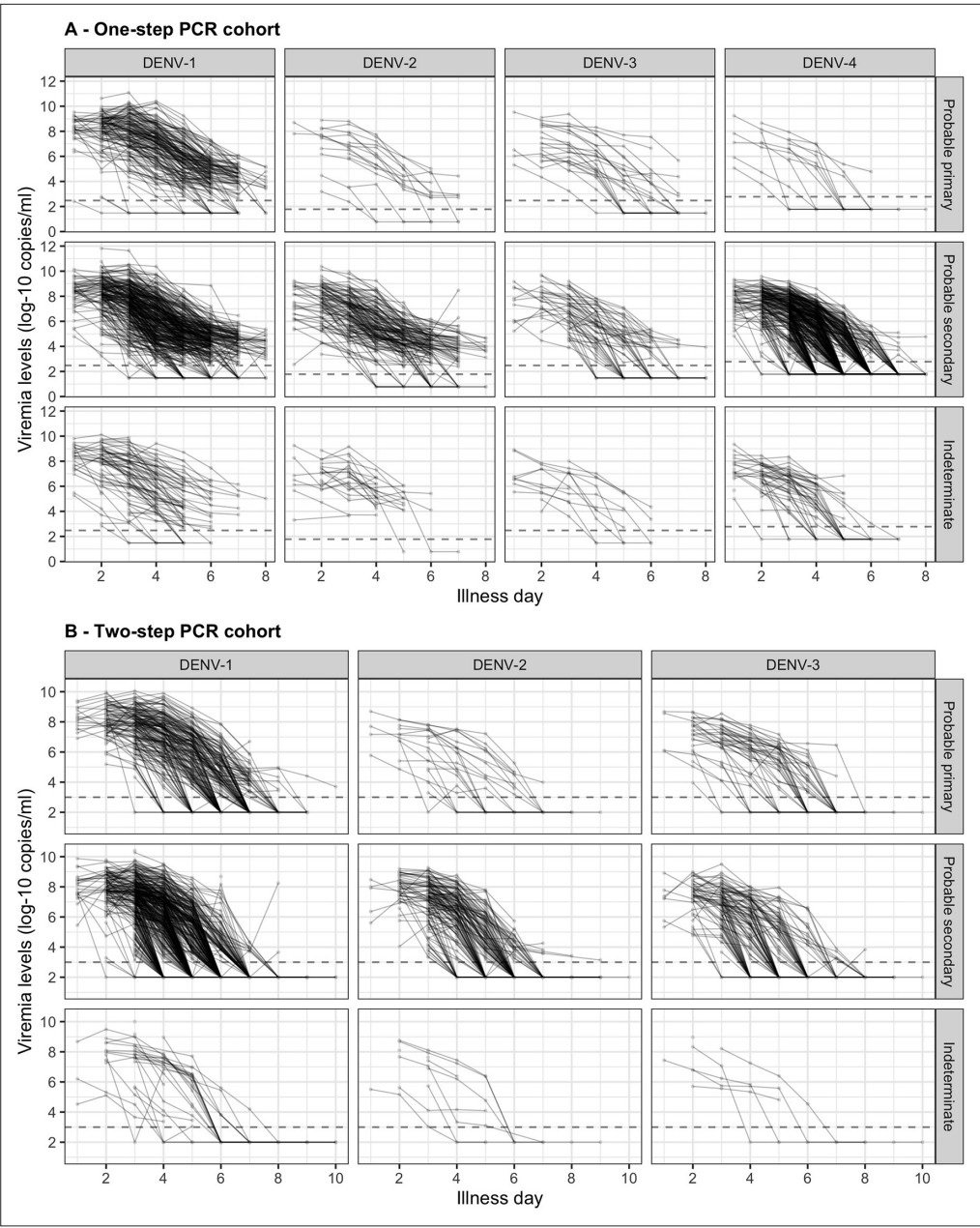

**Figure 2.** Individual trajectories of measured viremia levels for (**A**) the one-step PCR cohort and (**B**) the two-step PCR cohort. The dots represent the measured viremia levels and are connected by lines for each individual patient. The dashed lines indicate the detection limits, which are 300 copies/ml for DENV-1 and DENV-3, 60 copies/ml for DENV-2, and 600 copies/ml for DENV-4 in the one-step PCR cohort, and 1000 copies/ml in the two-step PCR cohort. Fifty-seven measured values that are lower than 1000 copies/ml in the two-step PCR cohort are considered as below the detection limit. Values below the detection limit are visually represented as 1/10 of the corresponding detection limit (on the original scale). DENV, dengue virus; PCR, polymerase chain reaction.

and illness day were all <0.0001 (*Appendix 5—table 1*). The sensitivity analysis using the 10th-root transformation of viremia gave results similar to the main analysis using log-10 viremia (*Appendix 5—figure 2*).

## Effect of viremia on subsequent platelet count

*Figure 4* shows the mean platelet counts by viremia levels based on the supermodel for the one-step PCR method. There was minimal impact of viremia on days 1–5 platelet count. However, higher

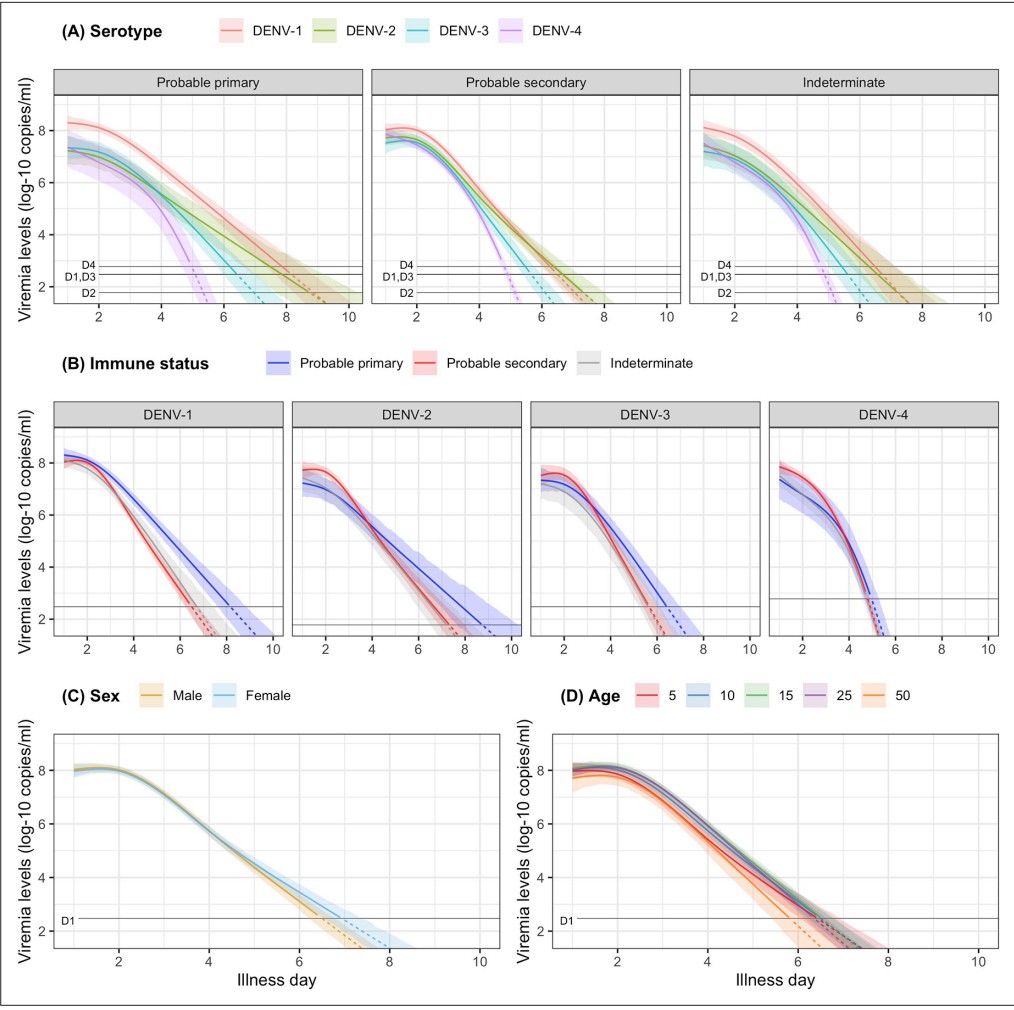

**Figure 3.** Fitted trends in mean viremia levels for (**A**) serotype, (**B**) immune status, (**C**) sex, and (**D**) age. The colored lines represent the estimated mean viremia levels, and the colored shaded regions represent the corresponding 95% credible intervals. The horizontal lines represent the detection limits (D1, D2, D3, and D4 denote DENV-1, DENV-2, DENV-3, and DENV-4, respectively). Dashed lines indicate fitted viremia levels that are below the detection limit. Viremia levels are shown for age of 10 years, male sex, serotype DENV-1, probable secondary infection, and using the one-step PCR. DENV, dengue virus; PCR, polymerase chain reaction.

viremia levels gave decreased subsequent platelet counts from day 6 onwards, for all serotypes. The strength of this effect increased with later illness day and was more pronounced in DENV-1 and DENV-2 compared to DENV-3 and DENV-4. All covariates had low p values (*Appendix 6—table 1*). The effect of viremia on subsequent platelet count remained consistent across subgroups of immune status, sex, and age. In the two-step PCR cohort, the effect of viremia on platelet count was less pronounced compared to the one-step PCR cohort, although the overall trends were similar (*Appendix 6—figure 1*).

## Effect of viremia on clinical outcomes

Higher viremia levels increased the risk of severe dengue and plasma leakage for each of the serotypes (*Figure 5*). The effect of viremia on both endpoints decreased with later landmark day of viremia measurement, particularly for severe dengue (*Appendix 7—table 1*). However, these trends were not clear in the results obtained from the simple logistic regression models at each landmark time point (*Appendix 7—figure 1*). This trend remained consistent across subgroups of sex, age, and study (*Appendix 7—figure 2*). Furthermore, the results were consistent between the analyses with and without imputation (*Appendix 7—figure 3*).

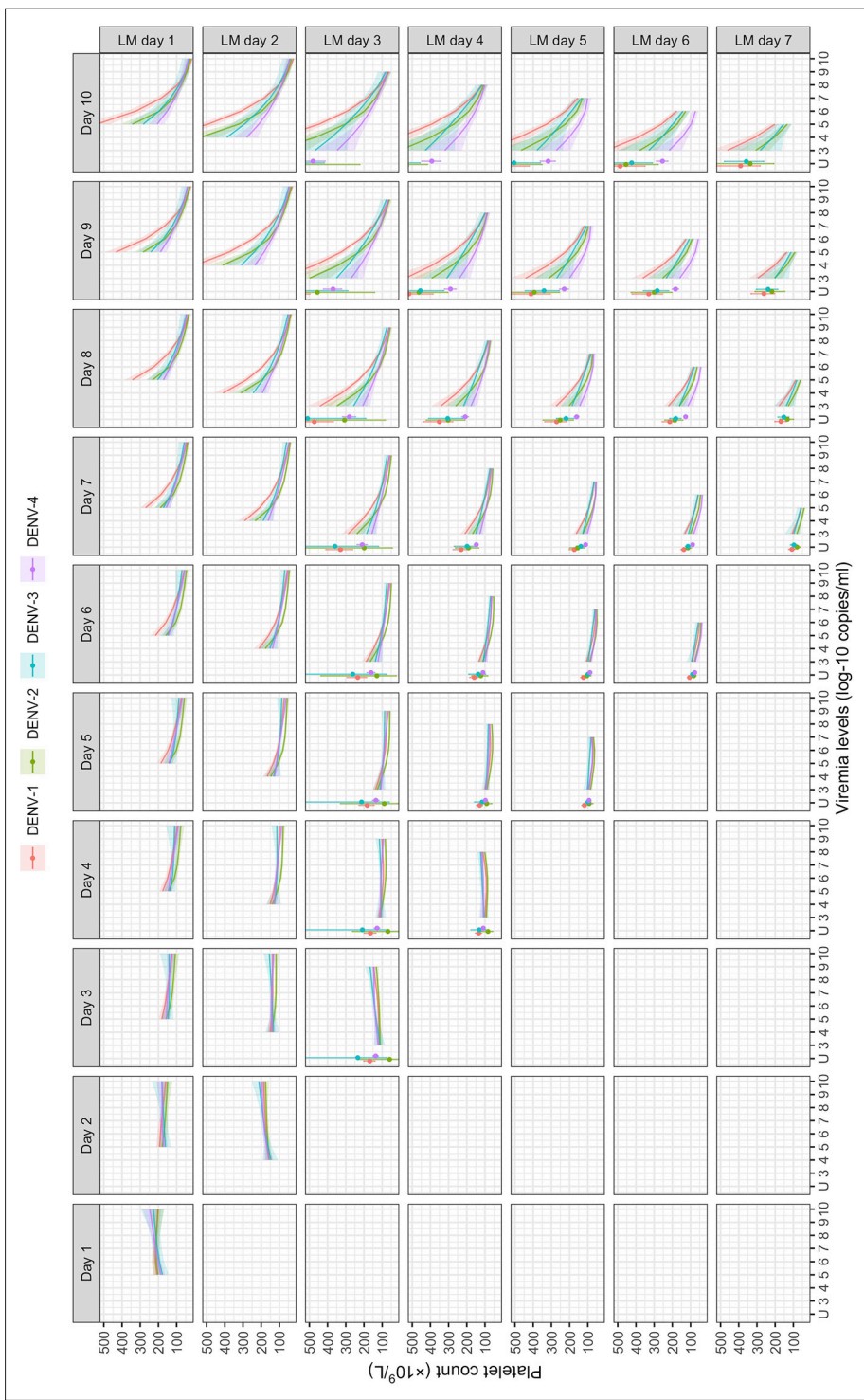

**Figure 4.** Fitted trends in mean platelet counts according to viremia levels – results from the supermodel. The colored lines or dots represent the estimated mean platelet counts. The colored shaded regions and whiskers indicate the corresponding 95% confidence intervals. Each row represents the effect of viremia on a specific day to platelet count from that day to day 10. No fitted trends are made for DENV-4 in LM day 7 since viremia was undetectable in almost all DENV-4 cases from day 7 onwards. The mean platelet counts are shown for age of 10 years, male sex, probable secondary infection, and using the one-step PCR. DENV, dengue virus; LM, landmark; PCR, polymerase chain reaction; U, under the limit of detection.

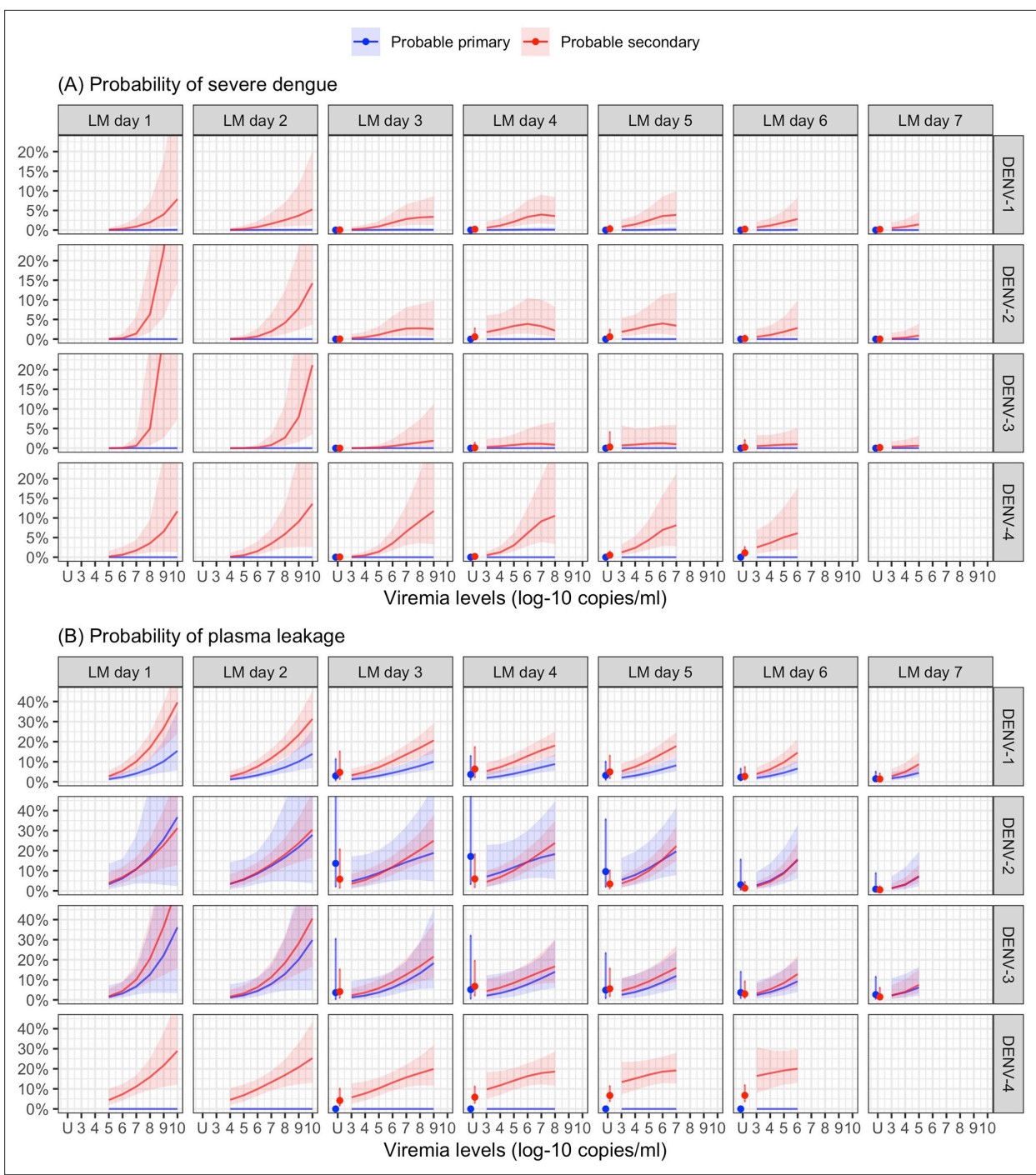

**Figure 5.** Probability of occurrence of (**A**) severe dengue and (**B**) plasma leakage, according to viremia levels – results from the supermodel. The colored lines or dots represent the probability of the endpoints. The colored shaded regions and whiskers indicate the corresponding 95% confidence intervals. Each column represents the effect of viremia on a specific day. No fitted trends are made for DENV-4 in LM day 7 since viremia was undetectable in almost all DENV-4 cases from day 7 onwards. Note that due to the limited number of severe dengue in primary infections (only 1 case in DENV-1) and plasma leakage in primary DENV-4 (see *Appendix 4—table 1*), the estimated probability of having these outcomes is nearly zero across all viremia levels within these subgroups. The probabilities are shown for age of 10 years, male sex, probable secondary infection, and from Study C. DENV, dengue virus, LM, landmark; PCR, polymerase chain reaction; U, under the limit of detection.

The supermodel revealed that older individuals had a relatively lower risk of developing severe dengue, but the trend was not significant. Males exhibited a slightly higher risk of severe dengue compared to females. The effect of serotype on the two endpoints was dependent on immune status. Individuals with probable secondary infection had a higher risk of experiencing both endpoints compared to those with probable primary infection (*Appendix 7—figure 4*).

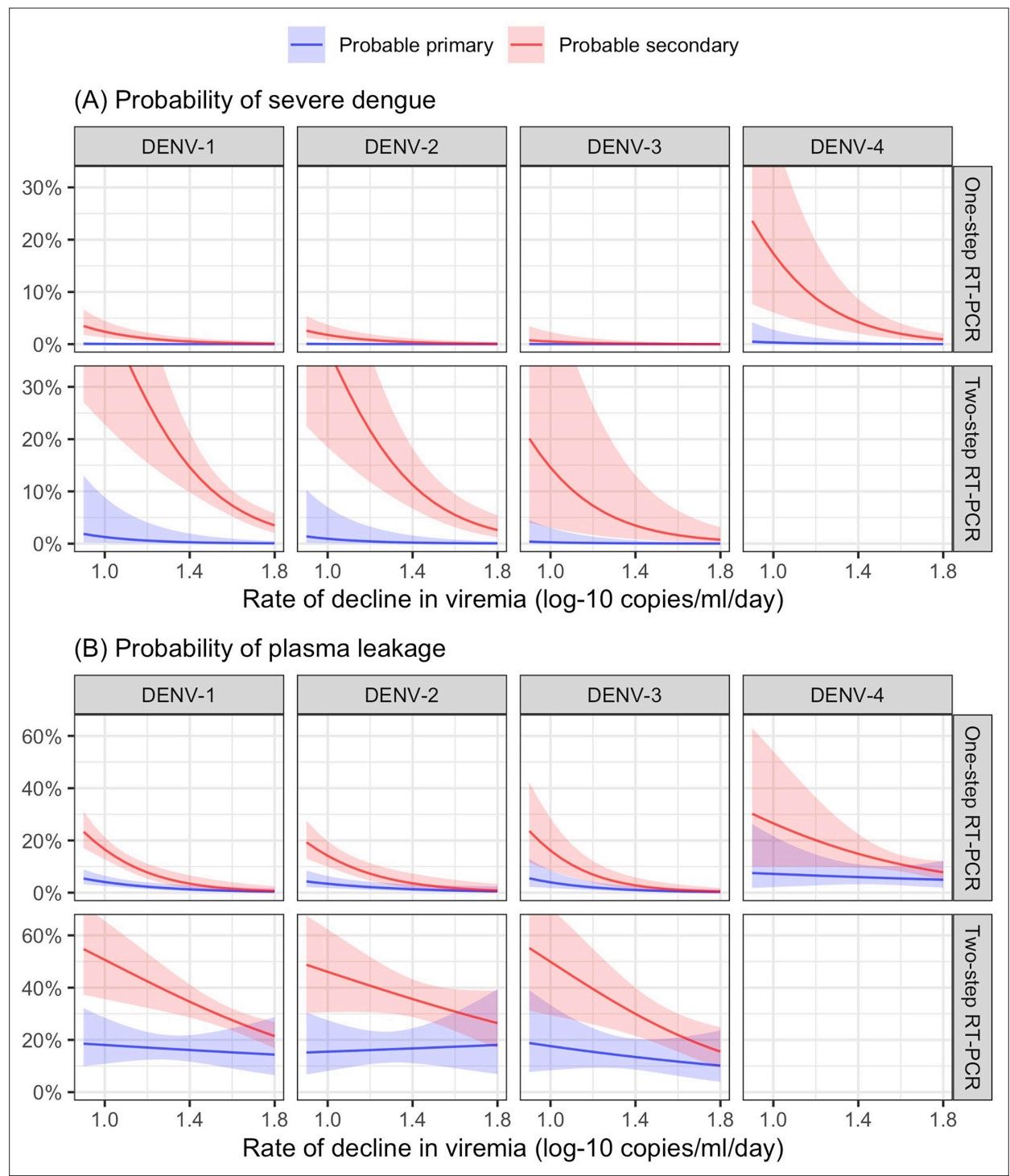

**Figure 6.** Probability of occurrence of (**A**) severe dengue and (**B**) plasma leakage, according to the rate of decline in viremia. The colored lines represent the probability of the endpoints. The colored shaded regions indicate the corresponding 95% confidence intervals.

From the model of viremia kinetics with a linear trend of illness day, the individual rate of viremia decline ranged from 0.58 to 2.01 log-10 copies/ml/day, with a median of 1.39 log-10 copies/ml/day. The logistic regression model demonstrated that a faster decline in viremia reduced the risk of severe dengue and plasma leakage (*Figure 6*). The effect on plasma leakage was stronger in the probable secondary infection group (*Appendix 7—table 2*).

## Discussion

In this study, we conducted a comprehensive analysis using a large dataset of 2340 viremic dengue patients to examine viremia kinetics from first presentation, its association with various virus and patient characteristics, and its impact on subsequent platelet count and disease severity. As others have described, we found that plasma viremia declines rapidly following the onset of symptoms, with the kinetics primarily influenced by the specific infecting serotype. Higher viremia levels were associated with a decrease in subsequent platelet count from day 6 onwards. Elevated viremia levels were found to increase the risk of developing severe dengue and plasma leakage, with the effect becoming weaker at later days. The effects of viremia on platelet count and clinical outcomes did not differ much by different subgroups of serotype, immune status, age, and sex. Moreover, a faster decline in viremia was associated with a reduced risk of the more severe clinical outcomes.

There is a limited number of published papers that studied the individual trajectory of dengue viremia. Most used data from Vietnam (*Ben-Shachar et al., 2016*; *Clapham et al., 2016*; *Clapham et al., 2014*; *Duyen et al., 2011*; *Nguyet et al., 2013*; *Simmons et al., 2007a*; *Tricou et al., 2011*), while one was from Thailand (*Matangkasombut et al., 2020*). Our study has provided further evidence supporting previous findings. We confirmed that (1) viremia rapidly decreases following the onset of symptoms, (2) DENV-1 exhibits higher viremia levels compared to DENV-2 and DENV-3, and (3) primary infection is associated with higher and more prolonged detectable viremia than secondary infection in DENV-1, while this pattern was not consistent across other serotypes. We have added viremia kinetics of DENV-4, and showed that viremia levels declined more rapidly than the other serotypes. Furthermore, our study demonstrated that the new PCR test has the ability to detect plasma viremia for a longer period compared to the older test. This may be attributed to the lower detection limits of the new test. Explaining the differences in viremia kinetics between serotypes remains challenging, as the molecular factors of the virus that influence plasma viremia levels are still unknown. It is also important to note that our study focused on the time since symptom onset rather than the time since infection. We cannot rule out that some of the observed differences are explained by a difference in time from infection to symptom onset.

Our study suggests that higher viremia levels on any day before day 6 are associated with reduced platelet count on days 6–8, typically corresponding to the nadir of platelet count. Additionally, it is possible that viremia in the initial 4 days after symptom onset could affect platelet count with a delay of 3–4 days. The direct involvement of the dengue virus in triggering platelet activation and apoptosis can contribute to the development of thrombocytopenia. Thrombocytopenia in dengue infection is thought to occur through two mechanisms: bone marrow suppression, which reduces thrombopoiesis, and increased peripheral platelet clearance (*Quirino-Teixeira et al., 2022*). Various processes contribute to these mechanisms, including platelet–leukocyte and platelet–endothelial cell interactions, phagocytosis, complement-mediated lysis, aggregation, and clot formation. Additionally, several host immune response factors are involved in platelet activation (*Balakrishna Pillai et al., 2022*). In our analysis, we considered interactions between viremia level and both serotype and immune status. Interestingly, we observed that the effect of viremia level on subsequent platelet count did not differ much by serotype and immune status.

Higher plasma viremia levels increase the risk of worse clinical outcomes, as demonstrated in our previous study that only utilized viremia at enrollment during the febrile phase (*Vuong et al., 2021*). The diminishing impact of viremia on the two endpoints on later illness days may be attributed to the heightened immune responses that are likely triggered by higher viremia levels, and the resulting complex interplay between these factors that underlies progression to severe dengue. The effects of viremia, age, serotype, immune status, and illness day on the clinical endpoints in this study are consistent with those observed in the previous study. The similarity between these two analyses, along with the weaker effect of viremia at later days, suggests that viremia levels around the time of symptom onset could serve as a reliable predictor of dengue severity. It suggests that the early febrile

phase, specifically illness days 1–3, is the critical period of measuring viremia levels in clinical practice and dengue studies. Secondary infection remains a substantial risk factor for more severe outcomes, while viremia kinetics are not influenced much by immune status. These findings suggest that immune status and viremia may be independent predictors of clinical outcomes, following distinct pathways.

The landmark approach enabled us to investigate the effect of viremia on platelet count and the clinical endpoints at each illness day, ranging from days 1 to 7, while ensuring that the viremia measurements preceded the occurrence of the outcomes. The supermodel allows for investigation of trends of the effects, while gaining power because we assume the effects of some fixed factors (e.g., age and sex) to remain constant over time. There might be other potential confounders such as host genetic and immune response factors, but these data were not available. Moreover, we assumed that the relation between viremia level and platelet count was one-way: viremia level affects platelet count. Whether platelet count affects viremia is unknown.

A notable strength of our study is the utilization of a large pooled dataset, which allowed for a flexible modeling approach to investigate the influence of age, sex, serotype, and immune status, as well as their interactions and nonlinear trends. The use of a model that accounts for left-censored data was needed for capturing the distribution of viremia levels, considering the significant proportion of undetectable values observed.

The study has several limitations. First, the study only included patients from Vietnam, which may restrict the generalizability of the findings to other countries and regions. Second, our study assessed viral RNA in the plasma, including both viable and non-viable viral particles, which could potentially lead to an overestimation of the infectious viral particles in the blood. Lastly, data during the incubation period were unavailable as patients are typically not diagnosed with dengue until symptom onset. Therefore, the trajectory of viremia from the time of infection onwards could not be demonstrated, and symptom onset was used as the reference point instead of infection. This could potentially lead to an inaccurate interpretation of the results if the incubation period is influenced by the same factors that we included in our analyses. For example, in cases of secondary infection, the immune response is stronger and quicker than in primary infection. This might lead to a shorter duration between infection and symptom onset, as well as faster viral clearance.

In conclusion, our findings reveal that viremia levels exhibit a rapid decline shortly after symptom onset, becoming undetectable after approximately 1 week in the majority of patients. Viremia kinetics display variations depending on the infecting serotype. Higher viremia levels are associated with a subsequent reduction in platelet count on illness days 6–8, and an increased risk of experiencing more severe clinical outcomes. Viremia serves as an important predictor of dengue outcomes, independent of the host immune status. However, when considering our previous analysis that involved single early viremia measurements, the addition of daily viremia measurements may not enhance the prediction of worse clinical outcomes. Thus, the measurement of viremia levels during the early febrile phase is an important marker for clinical practice and dengue-related research. Moreover, a faster decline in viremia was found to be associated with a reduced risk of more severe clinical outcomes, suggesting that viremia kinetics could be a good surrogate endpoint for phase-2 dengue therapeutic trials.

## Acknowledgements

We thank the staff from the Dengue group at the Oxford University Clinical Research Unit, Hospital for Tropical Diseases at Ho Chi Minh City (Vietnam) and the many hospitals and clinics who recruited and followed patients in the three studies. We gratefully acknowledge the patients and their relatives for participating in these studies.

## Additional information

### Competing interests

Thomas Jaenisch: reports receiving personal fees as members of the Roche Pharmaceuticals Advisory Board on Severe Dengue, outside the submitted work. Sophie Yacoub: reports receiving personal honorarium for attending the Novartis dengue drug ad board meeting and Takeda dengue education symposium, outside the submitted work. Bridget A Wills: reports receiving personal fees (a) as a

member of the Roche Advisory Board on Severe Dengue and (b) as a member of the Data Monitoring and Adjudication Committees for the Takeda dengue vaccine trials, both outside the remit of the submitted work. The other authors declare that no competing interests exist.

### Funding

| Funder | Grant reference number | Author |
| --- | --- | --- |
| Wellcome Trust | 084368/Z/07/Z | Cameron P Simmons |
| Wellcome Trust | 077078/Z/05/Z | Bridget A Wills |
| European Union Seventh Framework Programme | FP7-281803 IDAMS | Thomas Jaenisch |
| Wellcome Trust | 106680/Z/14/Z | Sophie Yacoub Ronald Geskus |
| Wellcome Trust | 077078/Z/05/A | Bridget A Wills |
| Wellcome Trust | 089276/B/09/Z | Bridget A Wills |
| Wellcome Trust | 225167/Z/22/Z | Sophie Yacoub Ronald Geskus |

The funders had no role in study design, data collection, and interpretation, or the decision to submit the work for publication. For the purpose of Open Access, the authors have applied a CC BY public copyright license to any Author Accepted Manuscript version arising from this submission.

### Author contributions

Nguyen Lam Vuong, Conceptualization, Data curation, Software, Formal analysis, Visualization, Methodology, Writing – original draft, Writing – review and editing; Nguyen Than Ha Quyen, Nguyen Thi Hanh Tien, Kien Duong Thi Hue, Huynh Thi Le Duyen, Resources, Investigation, Writing – review and editing; Phung Khanh Lam, Resources, Data curation, Supervision, Investigation, Methodology, Writing – review and editing; Dong Thi Hoai Tam, Tran Van Ngoc, Conceptualization, Resources, Investigation, Writing – review and editing; Thomas Jaenisch, Cameron P Simmons, Conceptualization, Funding acquisition, Investigation, Methodology, Writing – review and editing; Sophie Yacoub, Bridget A Wills, Ronald Geskus, Conceptualization, Supervision, Funding acquisition, Investigation, Methodology, Writing – review and editing

### Author ORCIDs

Nguyen Lam Vuong ![ORCID] http://orcid.org/0000-0003-2684-3041
Phung Khanh Lam ![ORCID] http://orcid.org/0000-0001-7968-473X
Cameron P Simmons ![ORCID] http://orcid.org/0000-0002-9039-7392
Bridget A Wills ![ORCID] https://orcid.org/0000-0001-9086-8804
Ronald Geskus ![ORCID] http://orcid.org/0000-0002-2740-3155

### Ethics

We used data from three prospective observational studies performed as part of the longstanding collaboration between the Oxford University Clinical Research Unit and the Hospital for Tropical Diseases in Ho Chi Minh City, Vietnam. All studies were approved by the Oxford Tropical Research Ethics Committee (OXTREC-012-05, OXTREC-017-02, and OXTRECT-40-11) and the Institutional Ethics Committee.

Reviewer #1 (Public review): https://doi.org/10.7554/eLife.92606.3.sa1
Reviewer #2 (Public review): https://doi.org/10.7554/eLife.92606.3.sa2
Author response https://doi.org/10.7554/eLife.92606.3.sa3

## Additional files

### Supplementary files
• MDAR checklist

## Data availability

All data generated or analyzed during this study and all codes have been deposited on GitHub at https://github.com/Nguyenlamvuong/Dengue_Viremia_Kinetics_eLife_2024 (copy archived at *Vuong, 2024* and https://doi.org/10.5281/zenodo.11243209).

The following dataset was generated:

| Author(s) | Year | Dataset title | Dataset URL | Database and Identifier |
| --- | --- | --- | --- | --- |
| Vuong NL | 2024 | Nguyenlamvuong/ Dengue_Viremia_ Kinetics_eLife_2024: Dengue_Viremia_Kinetics_ eLife_2024_with_data (dengue_viremia) | https://doi.org/ 10.5281/zenodo. 11243209 | Zenodo, 10.5281/ zenodo.11243209 |

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

# Appendix 1

## Protocol synopses for the three studies

The three studies described below were performed as part of the longstanding collaboration between the Oxford University Clinical Research Unit and the Hospital for Tropical Diseases in Ho Chi Minh City, Vietnam. All studies were approved by the Scientific and Ethics Committee of the hospital and by the Oxford Tropical Research Ethics Committee.

## Study A

### Title: A study of early dengue disease among children in the community

Study objectives: this study was part of a larger programme of work aiming to (1) estimate the burden of paediatric dengue disease in HCMC, (2) explore the mechanisms of the endothelial dysfunction associated with dengue by detailed study of well characterized patients throughout the evolution of the disease, and (3) investigate whether commonly available laboratory parameters (haematocrit, platelets, urine protein) measured during the febrile period can be used to predict subsequent disease severity.

Study period: 2000–2009

Study subjects: previously healthy children aged 5–15 years, presenting to hospital outpatient clinics or to one of several local health centers in Ho Chi Minh City on day 1 or 2 of a non-specific febrile illness. Children were followed daily in the community. In the event of admission to hospital, patients continued to be followed either on the dengue ward or on PICU of the Hospital for Tropical Diseases in HCMC, according to clinical severity.

Study setting: parents/guardians of eligible patients were informed about the study and gave consent. After enrollment baseline blood and urine samples were obtained. Patients were then asked to return each day for 7 days or until full recovery, and then at 1–2 weeks and 1–2 months for follow-up. The study nurse assessed each child in the clinic every day, obtained daily blood and urine samples, and requested a medical opinion if there were any concerns about the child or the daily blood results. Two trained study doctors supervised the nurses in the clinics and followed any patients admitted to the hospital. Management in the clinic or referral for hospital admission was at the discretion of the clinic doctors, in consultation with the dengue study doctors.

Data collection: demographic and clinical data were collected on each patient at study entry using a standard case report form. Clinical progress was documented each day specifically focusing on the occurrence of bleeding manifestations or signs of vascular leakage. There was no systematic data collection regarding organ dysfunction as the clinicians felt that these problems were rare in this age group, and no biochemistry (e.g., liver/renal function) test results were documented in the research files.

Laboratory confirmation of dengue: either positive reverse transcription polymerase chain reaction (RT-PCR) or seroconversion by ELISA (IgM or IgG or both). Testing for dengue non-structural protein 1 (NS1) was performed occasionally but was not part of the original study protocol.

Publications:

1. Dung NT, Duyen HT, Thuy NT, et al. Timing of CD8+ T cell responses in relation to commencement of capillary leakage in children with dengue. *J Immunol* 2010; **184**(12):7281–7287
2. Duyen HT, Ngoc TV, Ha do T, et al. Kinetics of plasma viremia and soluble nonstructural protein 1 concentrations in dengue: differential effects according to serotype and immune status. *J Infect Dis* 2011; **203**(9):1292–1300
3. Hanh Tien NT, Lam PK, Duyen HT, et al. Assessment of microalbuminuria for early diagnosis and risk prediction in dengue infections. *PLoS One* 2013; **8** (1):e54538

## Study B

### Title: Mild Dengue Study

Study objectives: this study was part of a larger programme of work aiming to (1) estimate the burden of pediatric dengue disease in HCMC, (2) explore the mechanisms of the endothelial dysfunction associated with dengue by detailed study of well characterized patients throughout the evolution of the disease, and (3) investigate whether commonly available laboratory parameters measured during the febrile period can be used to predict subsequent disease severity.

Study period: 2000–2009

Study subjects: children 5–15 years presenting to the Hospital for Tropical Diseases in HCMC with a febrile illness consistent with possible dengue, who required hospital admission to the paediatric dengue ward, NOT those admitted to HDU/ICU. Only children admitted from home were included (i.e., not transfers from other hospitals).

Study setting: parents/guardians of eligible patients were informed about the study and gave consent. All patients were followed daily until discharge with simple study notes and a daily full blood count (plus any other tests clinically indicated), and treated according to individual diagnoses. On the day of study enrolment all patients had a plasma sample obtained for research investigations. On the 6th illness day a second plasma sample was obtained. At discharge the notes were reviewed and each child was assigned a final diagnosis and clinical disease category, using carefully defined severity criteria for vascular leak and bleeding. The patients were asked to return for follow-up at 1 month, and again at 2 months if there were any ongoing concerns.

Data collection: demographic and clinical data were collected on each patient at study entry using a standard case report form. Clinical progress was documented each day specifically focusing on the occurrence of bleeding manifestations or signs of vascular leakage. There was no systematic data collection regarding organ dysfunction as the clinicians felt that these problems were rare in this age group, and no biochemistry (e.g., liver/renal function) test results were documented in the research files.

Laboratory confirmation of dengue: either positive RT-PCR or seroconversion by ELISA (IgM or IgG or both). NS1 testing was not performed.

Publications:

1. Dung NT, Duyen HT, Thuy NT, et al. Timing of CD8+ T cell responses in relation to commencement of capillary leakage in children with dengue. *J Immunol* 2010; **184**(12):7281–7287
2. Trung DT, Thao LTT, Dung NM, et al. Clinical Features of Dengue in a Large Vietnamese Cohort: Intrinsically Lower Platelet Counts and Greater Risk for Bleeding in Adults Than Children. *PLoS Negl Trop Dis* 2012; **6** (6):e1679.
3. Lam PK, Ngoc TV, Thuy TT, et al. The value of daily platelet counts for predicting dengue shock syndrome: Results from a prospective observational study of 2301 Vietnamese children with dengue. *PLoS Negl Trop Dis* 2017; **11**(4): e0005498
4. Hoang Quoc C, Henrik S, Isabel RB, et al. Synchrony of Dengue Incidence in Ho Chi Minh City and Bangkok. *PLoS Negl Trop Dis* 2016; **10**(12):e0005188

## Study C

### Title: IDAMS – observational study in early dengue (ClinicalTrials.gov Identifier: NCT01550016)

Study objectives: to improve diagnosis and clinical management of dengue through approaches designed (1) to differentiate between dengue and other common febrile illness within 72 hr of fever onset, and (2) among patients with dengue to identify markers predictive of the likelihood of evolving to a more severe disease course.

Study period: 2011–2016

Study subjects: both adults and children (≥5 years) were eligible for enrolment. This was a prospective multi-center observational study that enrolled approximately 7500 patients presenting with a febrile illness consistent with possible dengue to outpatient health facilities in urban centers in eight countries across Asia and Latin America. Following appropriate informed consent, subjects presenting at one of the designated sites with fever for ≤72 hr without localizing features, i.e., consistent with a possible diagnosis of dengue, were enrolled.

Study setting: following enrolment, clinical history and examination findings were recorded in the case report form and a 3- to 5-ml (age-dependent) research blood sample was obtained, together with appropriate samples to measure a range of haematological and biochemical parameters in line with local laboratory capacity. Patients were then reviewed daily in the OPD until fully recovered and afebrile for 24 hr, or for up to 6 days from enrolment. A full blood count was carried out each day, and on the last acute illness visit (within approximately 24 hr of defervescence) a second sample for a biochemical profile was obtained together with a sample for serology. All patients were then

asked to attend a final follow-up visit around days 10–14 of illness, at least 1 week from the last visit during the acute illness. All management decisions throughout the acute illness were at the discretion of the clinic physicians. Any patient requiring hospital admission continued to be followed daily using a similar but more detailed case report form (CRF), with the indication(s) for admission documented, and all management interventions recorded together with the physician's rationale for these interventions.

Data collection: a structured clinical questionnaire was completed upon enrolment and then once daily for up to 6 days for all patients in the study. This CRF included detailed clinical signs and symptoms, as well as all standard laboratory results including liver and renal function.

Laboratory confirmation of dengue: any case with virological evidence of dengue as shown by a positive RT-PCR assay or NS1 ELISA test, was defined as having laboratory-confirmed dengue.

Publications:

1. Nguyet MN, Duong TH, Trung VT, et al. Host and viral features of human dengue cases shape the population of infected and infectious Aedes aegypti mosquitoes. *Proc Natl Acad Sci USA* 2013; **110**(22):9072–9077
2. Jaenisch T, Tam DTH, Kieu NTT, et al. Clinical evaluation of dengue and identification of risk factors for severe disease: protocol for a multicentre study in 8 countries. *BMC Infect Dis* 2016; **16**: 120
3. Vuong NL, Le Duyen HT, Lam PK, et al. C-reactive protein as a potential biomarker for disease progression in dengue: a multi-country observational study. *BMC Med* 2020; 18(1): 35
4. Vuong NL, Lam PK, Ming DKY, et al. Combination of inflammatory and vascular markers in the febrile phase of dengue is associated with more severe outcomes. *eLife* 2021; 10: e67460

# Appendix 2

## Definitions used for dengue diagnostics and clinical severity classification

## Dengue diagnostics

Criteria for dengue diagnostics in the three studies were harmonized. Laboratory-confirmed dengue was defined by either a positive reverse transcription polymerase chain reaction (RT-PCR) or a positive dengue NS1 antigen test on plasma samples.

## Immune status

A probable primary infection was defined by two negative/equivocal IgG results on two samples taken at least 2 days apart during the first 10 days after symptom onset, with at least one sample obtained during the convalescent phase (illness days 6–10).

A probable secondary infection was defined by at least one positive IgG result during the first 10 days.

In cases without time-appropriate IgG results, immune status was classified as indeterminate.

## Clinical severity classification

| Outcome | Definition |
|---|---|
| Severe dengue | One or more of the following: <br> 1. Severe plasma leakage resulting in dengue shock syndrome and/or respiratory distress due to fluid accumulation <br> 2. Severe bleeding <br> 3. Severe organ impairment |
| | Dengue shock syndrome: Pulse pressure (difference between systolic and diastolic pressures) ≤20 mmHg or hypotension for age plus signs of poor capillary perfusion (cold extremities, delayed capillary refill, or rapid pulse rate). Clinician's assessment was accepted for occasional cases where the PP was between 20 and 25. |
| | Respiratory distress: Increased respiratory rate for age, with signs of increased work of breathing (retractions, nasal flaring, accessory muscle use) and need for additional support such as oxygen supplementation, continuous positive airway pressure (CPAP), or intubation. |
| Plasma leakage | 1. Hemoconcentration. Defined as ≥20% increase in hematocrit (HCT) from baseline (minimum HCT within illness days 1–3) to acute phase (maximum HCT within illness days 4–7) <br> 2. And/or evidence of fluid accumulation (pleural/peritoneal) on ultrasound or X-ray |
| | Severe: Dengue shock syndrome and/or respiratory distress due to plasma leakage |
| | Moderate: Evidence of plasma leakage but never developed shock or respiratory distress |
| | None: Hemoconcentration <20% (definition as above) and no evidence of fluid accumulation if chest X-ray/ultrasound done |
| | Indeterminate: Missing data for HCT (either baseline or acute phase or both) |
| Severe bleeding | One or more of the following: <br> 1. Any bleeding leading to hemodynamic instability <br> 2. Any bleeding resulting in death or permanent disability <br> 3. Any bleeding into a critical organ (e.g., central nervous system bleed) <br> 4. Any bleeding that results in need for blood transfusion <br> 5. Any bleeding that persists after measures are taken to stop bleeding (e.g., application of pressure) AND patient requires more intensive monitoring in an ICU or HDU |

*Continued on next page*

*Continued*

| Outcome | Definition |
|---|---|
| Severe organ impairment | Assessments were not systematic. For patients investigated on the basis of the attending clinician's concerns the following definitions were applied. |
| | Severe liver involvement: An acute clinical syndrome consistent with acute hepatitis, with new onset jaundice or coagulopathy (INR ≥1.5) or encephalopathy. |
| | Severe neurological involvement: Any new onset acute neurological signs or symptoms, except occurrence of a single simple febrile convulsion with full recovery within 30 min. |
| | Severe renal involvement: Increase in creatinine to more than 1.5 times the upper limit of normal for age, without existing kidney disease. |

Note that the criteria used for classifying dengue severity are derived from the World Health Organization guidelines (*World Health Organization, 2009*) and standard endpoint definitions for dengue trials (*Tomashek et al., 2018*). These criteria have been adjusted to align with the actual circumstances and the data accessible for the studies included in this analysis.

## Appendix 3

### Statistical analysis

We constructed a directed acyclic graph (DAG) to outline the hypothesized pathways from viremia level to platelet count and clinical outcomes.

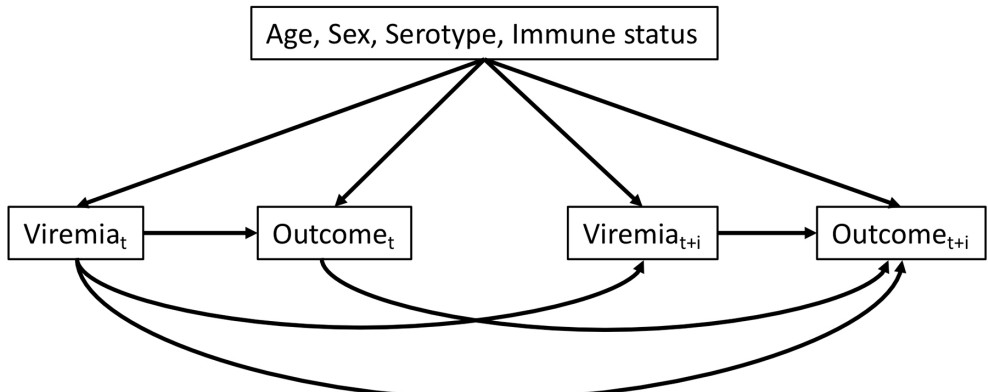

**Appendix 3—figure 1.** Directed acyclic graph illustrating the presumed causal relationships among variables. *t: illness day t (from days 1 to 7). t + i: after day t (i ≥ 1). Outcome can be platelet count or clinical outcomes (severe dengue or plasma leakage)*

### Analysis #1: Viremia kinetics and association with clinical characteristics

In the one-step PCR cohort, viremia levels were left-censored at specific detection limits: 300 copies/ml for DENV-1 and DENV-3, 60 copies/ml for DENV-2, and 600 copies/ml for DENV-4.

For the two-step PCR cohort, viremia levels were left-censored at 1000 copies/ml for DENV-1, DENV-2, and DENV-3. Due to the limited number of cases (two patients), DENV-4 was excluded from this cohort. We chose 1000 copies/ml as the detection limit for the two-step PCR method (the old method) based on assumptions of its higher detection limit compared to the one-step PCR method (the new method) which has been validated.

To account for nonlinear trends, splines were used for age (with three knots at 7, 13, and 28 years) and illness day (with four knots at days 1, 2, 4, and 6).

To fit the linear mixed-effects models with censored outcome, a Bayesian framework using Markov chain Monte Carlo (MCMC) methods was implemented using the R package 'MCMCglmm'. The models were run for a total of 20,000 MCMC iterations. Model adequacy was assessed through parameter trace plots and empirical distribution estimation (*Hartig, 2021*) from the models, indicating satisfactory fit for both models for log-10 viremia and 10th-root viremia.

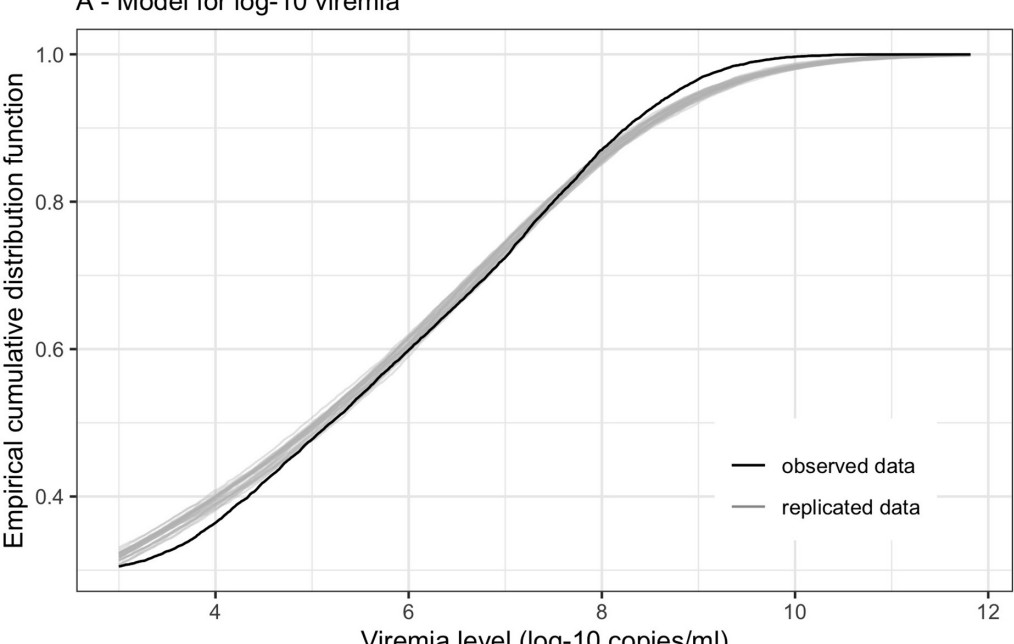

**Appendix 3—figure 2.** Empirical distribution estimation obtained from viremia kinetics model for (**A**) log-10 viremia and (**B**) 10th-root viremia. The figures show the empirical cumulative distribution function of the observed viremia levels (represented by the black curve), along with the empirical cumulative distribution function derived from 100 simulated sets of viremia levels generated by the model (represented by the gray curves). The plots are started from 1000 copies/ml (on the original scale). The close resemblance between the observed and simulated distributions in both models based on log-10 and 10th-root transformations of viremia levels suggests that the models provide a relatively good fit to the data.

The model using 10th-root viremia showed a relatively better fit, particularly for high viremia levels. However, considering the common use of the log transformation of viremia in studies, and the similarity of the predictions between the two models, we decided to use log-10 viremia as the main analysis of this study.

## Analysis #2: Effect of viremia on platelet count

### Analysis #2a: Compare the effect of viremia level on platelet count on illness days 6–8 using linear mixed-effects models. Separate models were fitted for viremia level measured 0–5 days earlier than the platelet count

We investigated the effect on platelet count of viremia 0–5 days earlier. Since almost all viremia values were undetectable after day 8, data for this analysis was restricted to days 6–8 and included patients who had available data on viremia on the same day and viremia 1–5 days earlier. We used a linear mixed-effects model for platelet count, including as covariates log-10 viremia, age, sex, serotype, immune status, PCR method, and illness day. We included interactions between illness day and all other covariates, interactions between log-10 viremia level and serotype and immune status, and interaction between serotype and immune status. Potential nonlinear effects of log-10 viremia and age were investigated using splines with three knots at the 10th, 50th, and 90th percentiles. To assess model fit, we examined the log-likelihood, Akaike information criterion (AIC), and Bayesian information criterion (BIC) values for each model (*Appendix 3—table 1*). Among the models considered, the model for viremia 3 days earlier demonstrated the best fit.

**Appendix 3—table 1.** Comparison of the models in Analysis #2a.

| Model | Degrees of freedom | Log-likelihood | AIC | BIC |
|---|---|---|---|---|
| Viremia on the same day | 55 | −502.0 | 1114.1 | 1384.3 |
| Viremia 1 day earlier | 55 | −488.2 | 1086.3 | 1356.6 |
| Viremia 2 days earlier | 55 | −479.0 | 1068.0 | 1338.3 |
| Viremia 3 days earlier | 55 | −452.4 | 1014.9 | 1285.1 |
| Viremia 4 days earlier | 55 | −469.6 | 1049.3 | 1319.5 |
| Viremia 5 days earlier* | 54 | −478.4 | 1064.8 | 1330.1 |

*The model for viremia 5 days earlier had fewer degrees of freedom due to the absence of a binary variable indicating detectable or undetectable viremia values (as all patients had detectable viremia values 5 days earlier).

### Analysis #2b: Compare the effect of viremia level on platelet count using linear regression. Separate models were fitted for platelet count on illness days 2–8 and for viremia level measured 0–5 days tearlier than the platelet count

To validate the findings from Analysis #2a, we conducted standard linear regression models to assess the relation between viremia level and platelet count for each combination of platelet count on illness days 2–8 and viremia level on days 0–5 before the platelet measurement. The models included log-10 viremia (with splines using three knots at the 10th, 50th, and 90th percentiles), the binary variable indicating detectable or undetectable viremia (if applicable), age (with splines using three knots at 7, 13, and 28 years), sex, serotype, immune status, and PCR method as covariates. No interaction terms were included due to limited data availability for each illness day. To evaluate model fit, we assessed the log-likelihood value for each model (*Appendix 3—table 2*).

**Appendix 3—table 2.** Comparison of the models in Analysis #2b.

| Model | Day 2 | Day 3 | Day 4 | Day 5 | Day 6 | Day 7 | Day 8 |
|---|---|---|---|---|---|---|---|
| Same day | 2.0 (24) | −58.7 (145) | −95.9 (220) | −99.7 (227) | −84.9 (198) | −252.6 (533) | −210.5 (449) |
| 1 day earlier | 0.9 (24) | **−53.1 (134)** | −95.2 (218) | −98.0 (224) | −80.1 (188) | −243.4 (515) | −208.7 (443) |
| 2 days earlier | | −56.0 (138) | **−88.8 (206)** | −94.0 (216) | −80.9 (190) | −232.8 (494) | −205.2 (438) |
| 3 days earlier | | | −91.8 (210) | **−89.8 (208)** | **−76.2 (180)** | **−229.3 (487)** | **−189.6 (407)** |
| 4 days earlier | | | | −92.9 (212) | −78.7 (185) | −241.3 (511) | **−189.6 (407)** |
| 5 days earlier | | | | | −83.0 (192) | −245.4 (517) | −196.5 (419) |

Statistics are log-likelihood (AIC). The highest log-likelihood values (along with the lowest AIC) per column are highlighted in bold. Note that all analyses per each column use the same sample size.

Overall, the models for viremia level 3 days earlier performed the best. These models had the highest log-likelihood values on illness days 5–8, and the second-highest log-likelihood values on illness day 4, with values closely approaching the highest log-likelihood values. These results align with the analyses conducted in Analysis #2a.

## Analysis #2c: Effect of days 1–7 viremia on platelet count on days 1–10

The formula of the supermodel was as follows:

$$\text{Fixed effect} : \text{PLT} \sim (\text{Viremia} + \text{LOD}) * \text{Serotype} * (\text{Immune} + \text{PCR}) + \text{Age} + \text{Sex} + \text{DOI} + \text{LM} +$$

$$\text{DOI} : [(\text{Viremia} + \text{LOD}) * \text{Serotype} * \text{PCR} + \text{Immune} + \text{Age} + \text{Sex} + \text{LM}] +$$

$$\text{LM} : [(\text{Viremia} + \text{LOD}) * \text{Serotype} * \text{PCR}]$$

$$\text{Random effect} : \text{PLT} \sim (\text{DOI} \mid \text{id} : \text{LM})$$

- PLT: platelet count
- Viremia: splines of log-10 viremia with three knots at 4, 6, and 8
- LOD: the binary variable indicating detectable or undetectable viremia
- Immune: immune status with three levels
- PCR: PCR method (one- or two-step)
- Age: splines of age with three knots at 7, 13, and 28
- DOI: splines of illness day with three knots at 2, 5, and 8
- LM: splines of landmark day with three knots at 2, 4, and 6, except for in the random effect, landmark day was treated as a factor variable with seven levels

The model was well fitted, demonstrated by the empirical distribution estimation plot generated from the model.

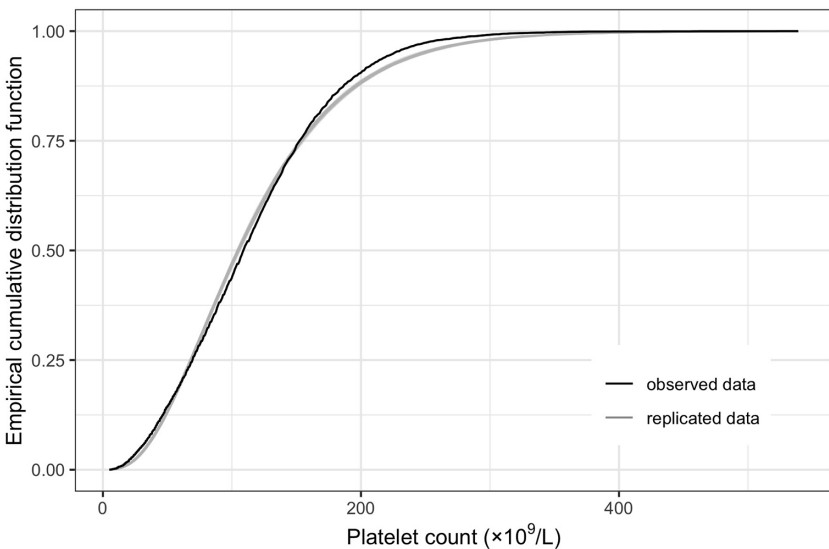

**Appendix 3—figure 3.** Empirical distribution estimation obtained from platelet count model. The figure shows the empirical cumulative distribution function of the observed platelet counts (represented by the black curve), along with the empirical cumulative distribution function derived from 100 simulated data points generated by the model (represented by the gray curves). The close resemblance between the observed and simulated distributions suggests that the model provides a relatively good fit to the data.

## Analysis #3: Effect of viremia level on clinical outcomes

The unknown day of plasma leakage for 26 patients was set at day 5.

For the supermodel that combined all landmark datasets, we employed the 'independence' covariance structure for the generalized estimating equations. The covariates included log-10 viremia level, the binary variable describing detectable or undetectable viremia, age, sex, serotype, immune status, study, and landmark illness day. Splines were used to allow for nonlinear trends by

log-10 viremia, age, and landmark day. We included the interactions between viremia with all other covariates (age, sex, and study) and the four-way interaction between log-10 viremia with serotype, immune status, and landmark day.

Due to the limited number of severe dengue cases (65 in total), the interactions allowed in the models for severe dengue and plasma leakage differed:

For severe dengue, we did not include any interactions between the binary variable describing detectable viremia with other covariates. Log-10 viremia and landmark day were specified as a linear trend in the interactions.

For plasma leakage, we additionally included the interactions between the binary variable describing detectable viremia with all other covariates (age [as a linear trend], sex, serotype, immune status, study, and landmark illness day [as a linear trend]). In the interactions between log-10 viremia and others, log-10 viremia was specified as a linear trend, whereas age and landmark day was specified with splines (similar to in the main terms).

The formula of the supermodels was as follows:

$$\text{Severe dengue} \sim \text{Viremia}_{[\text{splines}]} + \text{LOD} + \text{LM}_{[\text{splines}]} + \text{Serotype} * \text{Immune Age}_{[\text{splines}]} + \text{Sex} + \text{Study}+$$
$$\text{Viremia}_{[\text{linear}]} : (\text{LM}_{[\text{splines}]} * \text{Serotype} * \text{Immune} + \text{Age}_{[\text{splines}]} + \text{Sex} + \text{Study})$$

$$\text{Plasma leakage} \sim \text{Viremia}_{[\text{splines}]} + \text{LOD} + \text{LM}_{[\text{splines}]} + \text{Serotype} * \text{Immune} + \text{Age}_{[\text{splines}]} + \text{Sex} + \text{Study}-$$
$$\text{Viremia}_{[\text{linear}]} : (\text{LM}_{[\text{splines}]} * \text{Serotype} * \text{Immune} + \text{Age}_{[\text{linear}]} + \text{Sex} + \text{Study})+$$
$$\text{LOD} : (\text{LM}_{[\text{Linear}]} + \text{Sex} + \text{Study})$$

- Viremia$_{[\text{splines}]}$: splines of log-10 viremia with three knots at 4, 6.4, and 8.5
- Viremia$_{[\text{linear}]}$: linear trend of log-10 viremia
- LOD: the binary variable indicating detectable or undetectable viremia
- Immune: immune status
- Age$_{[\text{splines}]}$: splines of age with three knots at 7, 13, and 28
- Age$_{[\text{linear}]}$: linear trend of age
- LM$_{[\text{splines}]}$: splines of landmark day with three knots at 2, 4, and 6
- LM$_{[\text{linear}]}$: linear trend of landmark day

Some missing data were present for plasma leakage and immune status (i.e., the indeterminate immune status). In the complete-case analysis, individuals with missing plasma leakage were excluded, and immune status was analyzed with three groups, including indeterminate status. In the imputed data analysis, we used results from the imputation performed in the previous analysis of the effect of viremia in the early febrile phase on clinical outcomes (*Vuong et al., 2021*). Logistic regression was the imputation model for immune status and plasma leakage. All available baseline data were included in the imputation model, including study, age, gender, weight, enrolment date, illness day at enrolment, serotype, immune status, hematocrit and platelet count at enrolment, and the outcomes (hospitalization along with its characteristics [illness day when hospitalized and length of hospital stay], plasma leakage, and severe dengue). Twenty imputed datasets were generated, and 25 cycles per dataset were performed. Rubin's rules were applied to pool estimates from the logistic regression models over the imputed datasets.

# Appendix 4

## Descriptive analysis

**A - Log-10 transformation**

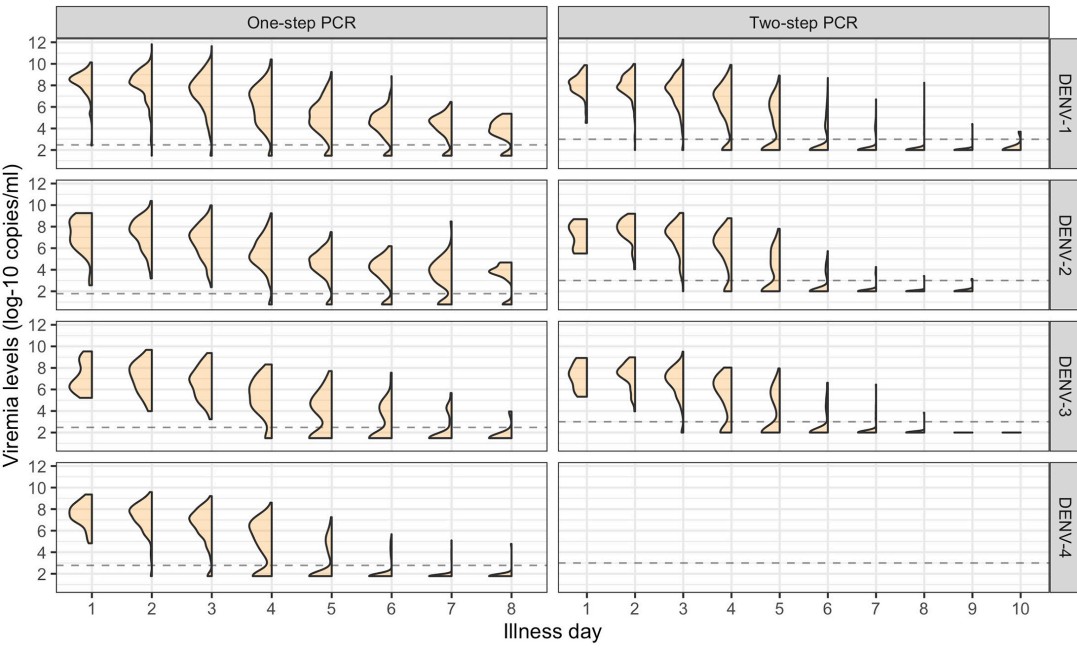

**B - 10th-root transformation**

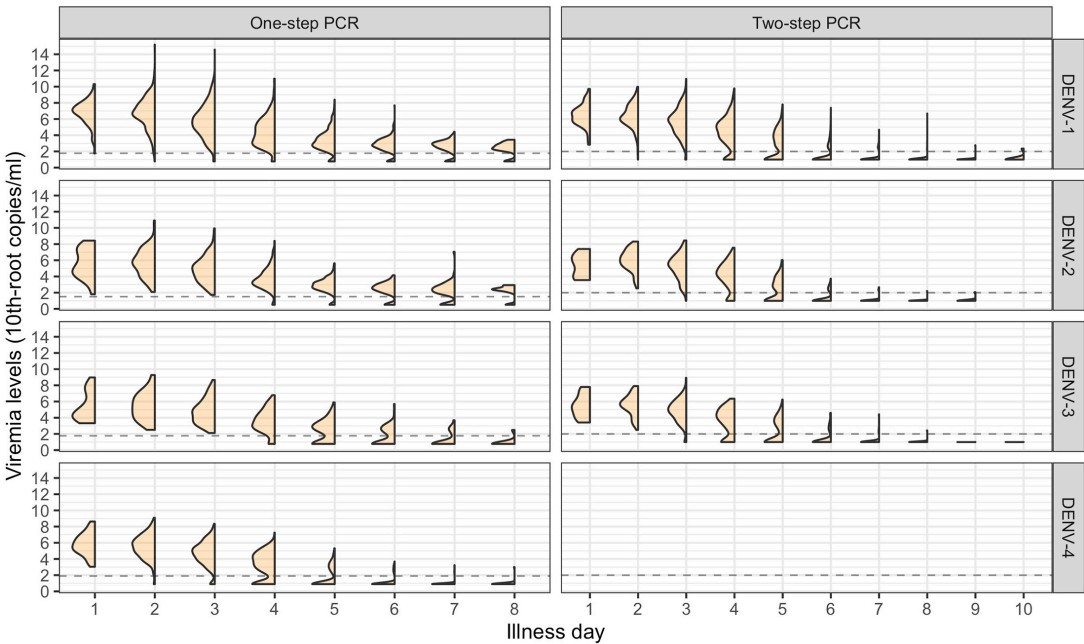

**Appendix 4—figure 1.** Distribution of (**A**) log-10 and (**B**) 10th-root viremia levels by illness day. Values below the detection limit are set as 1/2 of the detection limit in each corresponding transformation scale.

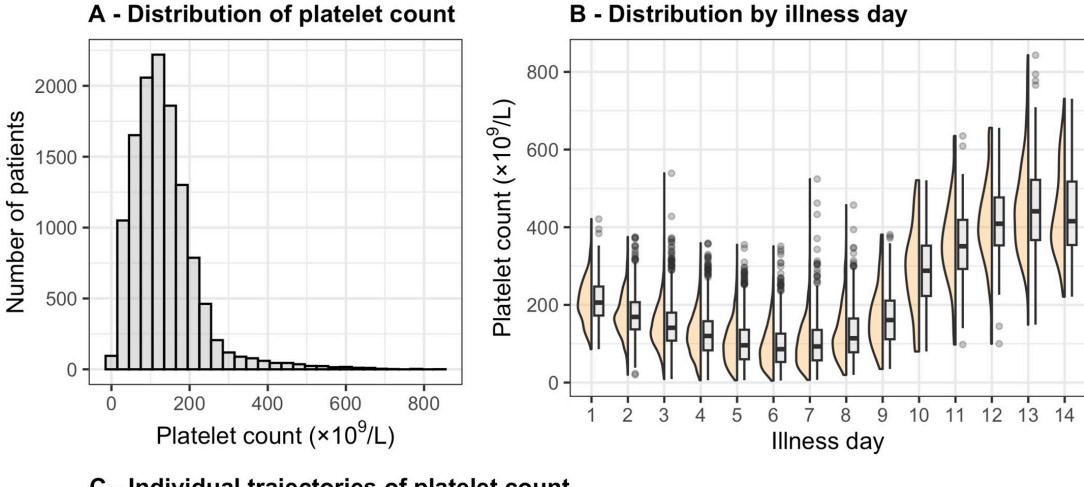

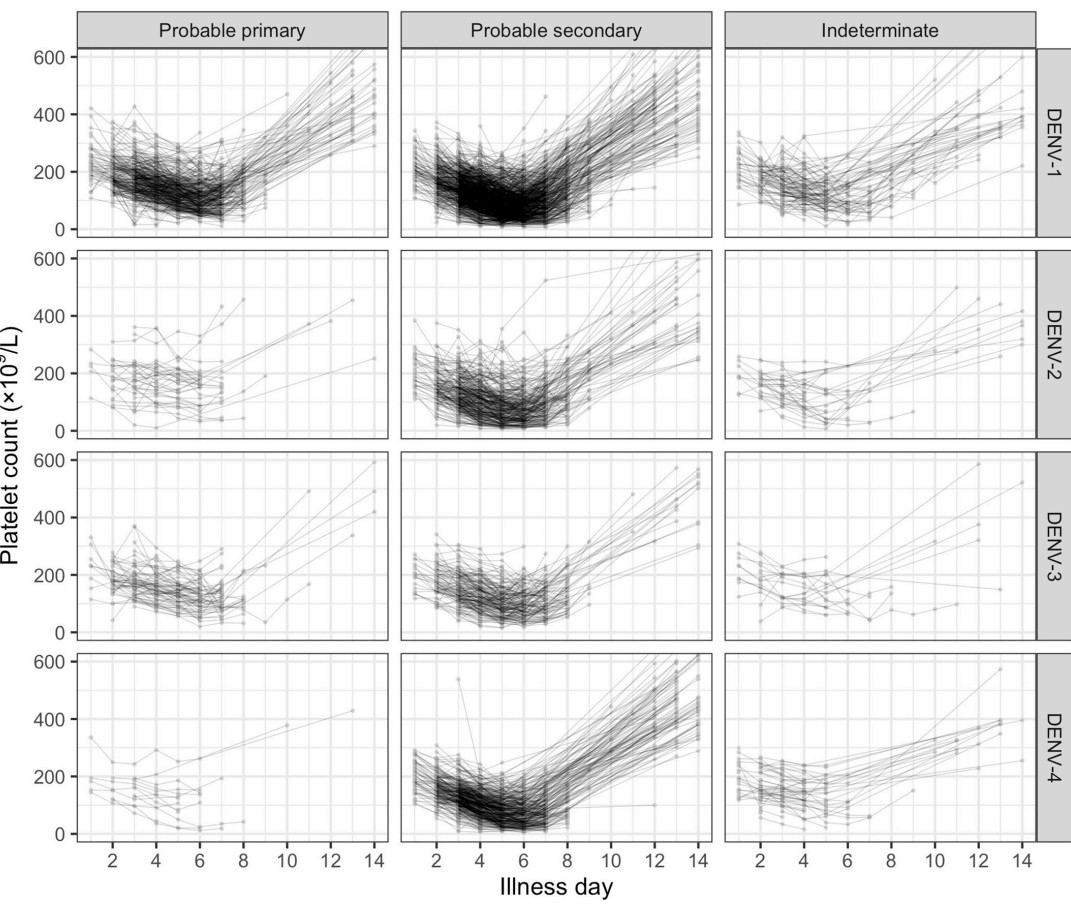

**Appendix 4—figure 2.** Distribution and individual trajectory of platelet count. The distribution of platelet count is shown with all measurements (**A**) and by illness day (**B**). In (**B**), the line inside each box represents the median, the upper and lower margins of each box indicate the interquartile range (25th; 75th percentile), and the violin curves illustrate the distribution of platelet count. Individual trajectories of platelet count are shown in (**C**): the dots represent the measured platelet counts and are connected by lines for each individual patient. The summary is restricted up to illness day 14. DENV, dengue virus.

**Appendix 4—table 1.** Distribution of clinical outcomes by serotype and immune status.

| | | Severe dengue* | | Plasma leakage* | | |
| | N†† | No (n = 2275) | Yes (n = 65) | No (n = 1935) | Yes (n = 353) | Indeterminate (n = 52) |
| --- | --- | --- | --- | --- | --- | --- |
| **DENV-1** | **1264** | **1224 (96.8)** | **40 (3.2)** | **1028 (81.3)** | **200 (15.8)** | **36 (2.8)** |
| Probable primary | 356 (28.2) | 355 (99.7) | 1 (0.3) | 307 (86.2) | 35 (9.8) | 14 (3.9) |
| Probable secondary | 774 (61.2) | 737 (95.2) | 37 (4.8) | 603 (77.9) | 151 (19.5) | 20 (2.6) |
| Indeterminate | 134 (10.6) | 132 (98.5) | 2 (1.5) | 118 (88.1) | 14 (10.4) | 2 (1.5) |
| **DENV-2** | **373** | **358 (96.0)** | **15 (4.0)** | **292 (78.3)** | **75 (20.1)** | **6 (1.6)** |
| Probable primary | 37 (9.9) | 37 (100.0) | 0 (0.0) | 29 (78.4) | 7 (18.9) | 1 (2.7) |
| Probable secondary | 299 (80.2) | 286 (95.7) | 13 (4.3) | 232 (77.6) | 63 (21.1) | 4 (1.3) |
| Indeterminate | 37 (9.9) | 35 (94.6) | 2 (5.4) | 31 (83.8) | 5 (13.5) | 1 (2.7) |
| **DENV-3** | **252** | **250 (99.2)** | **2 (0.8)** | **208 (82.5)** | **34 (13.5)** | **10 (4.0)** |
| Probable primary | 68 (27.0) | 68 (100.0) | 0 (0.0) | 58 (85.3) | 8 (11.8) | 2 (2.9) |
| Probable secondary | 165 (65.5) | 163 (98.8) | 2 (1.2) | 132 (80.0) | 25 (15.2) | 8 (4.8) |
| Indeterminate | 19 (7.5) | 19 (100.0) | 0 (0.0) | 18 (94.7) | 1 (5.3) | 0 (0.0) |
| **DENV-4** | **451** | **443 (98.2)** | **8 (1.8)** | **407 (90.2)** | **44 (9.8)** | **0 (0.0)** |
| Probable primary | 13 (2.9) | 13 (100.0) | 0 (0.0) | 13 (100.0) | 0 (0.0) | 0 (0.0) |
| Probable secondary | 381 (84.5) | 373 (97.9) | 8 (2.1) | 337 (88.5) | 44 (11.5) | 0 (0.0) |
| Indeterminate | 57 (12.6) | 57 (100.0) | 0 (0.0) | 57 (100.0) | 0 (0.0) | 0 (0.0) |

Summary statistics are n (%).

DENV: dengue virus.

*The percentages are computed by the total sample size per row.

†The percentages of immune status are computed by the total sample size within each DENV serotype.

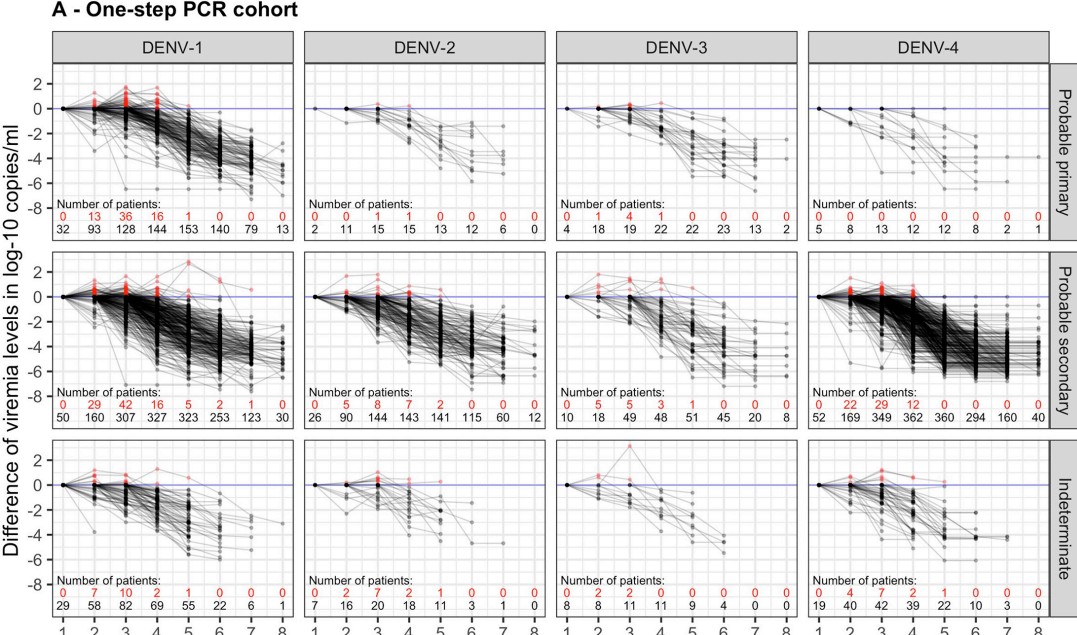

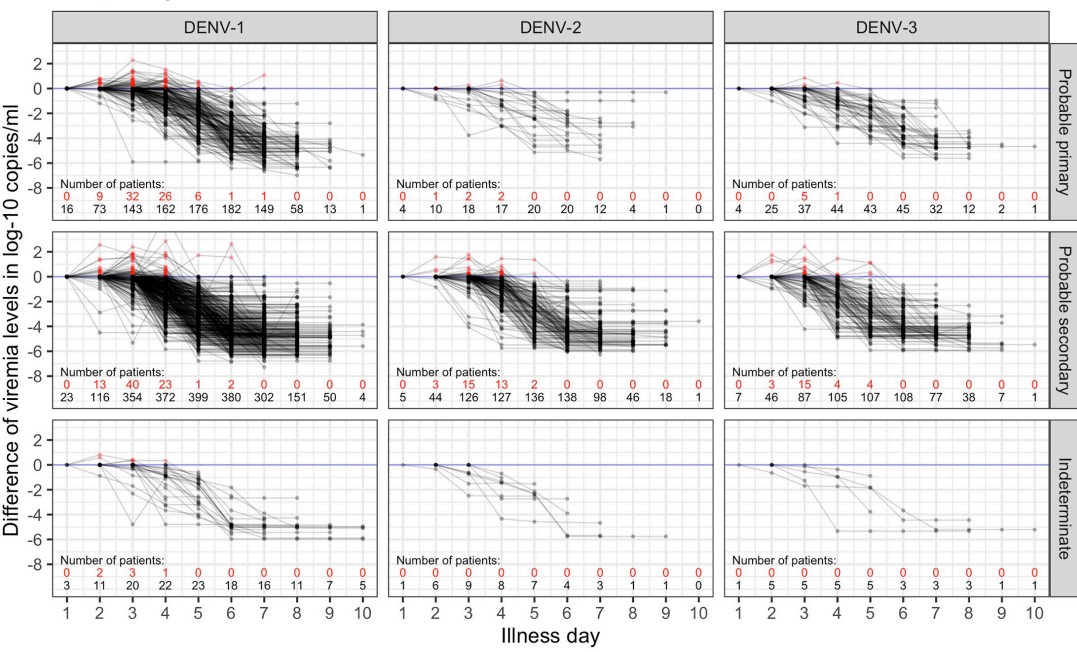

**Appendix 4—figure 3.** Difference in log-10 viremia from first measurement for (**A**) the one-step PCR cohort and (**B**) the two-step PCR cohort. The red dots represent values that are higher than the first value. The red and black numbers indicate the number of measurements that are higher and lower than the first value, respectively. The upper limit of the y-axis is set at 2.5 to provide a clearer visualization. Values below the detection limit are set as the corresponding detection limit to calculate the differences. DENV, dengue virus; PCR, polymerase chain reaction.

## Appendix 5

## Results for viremia kinetics and relationship with clinical characteristics

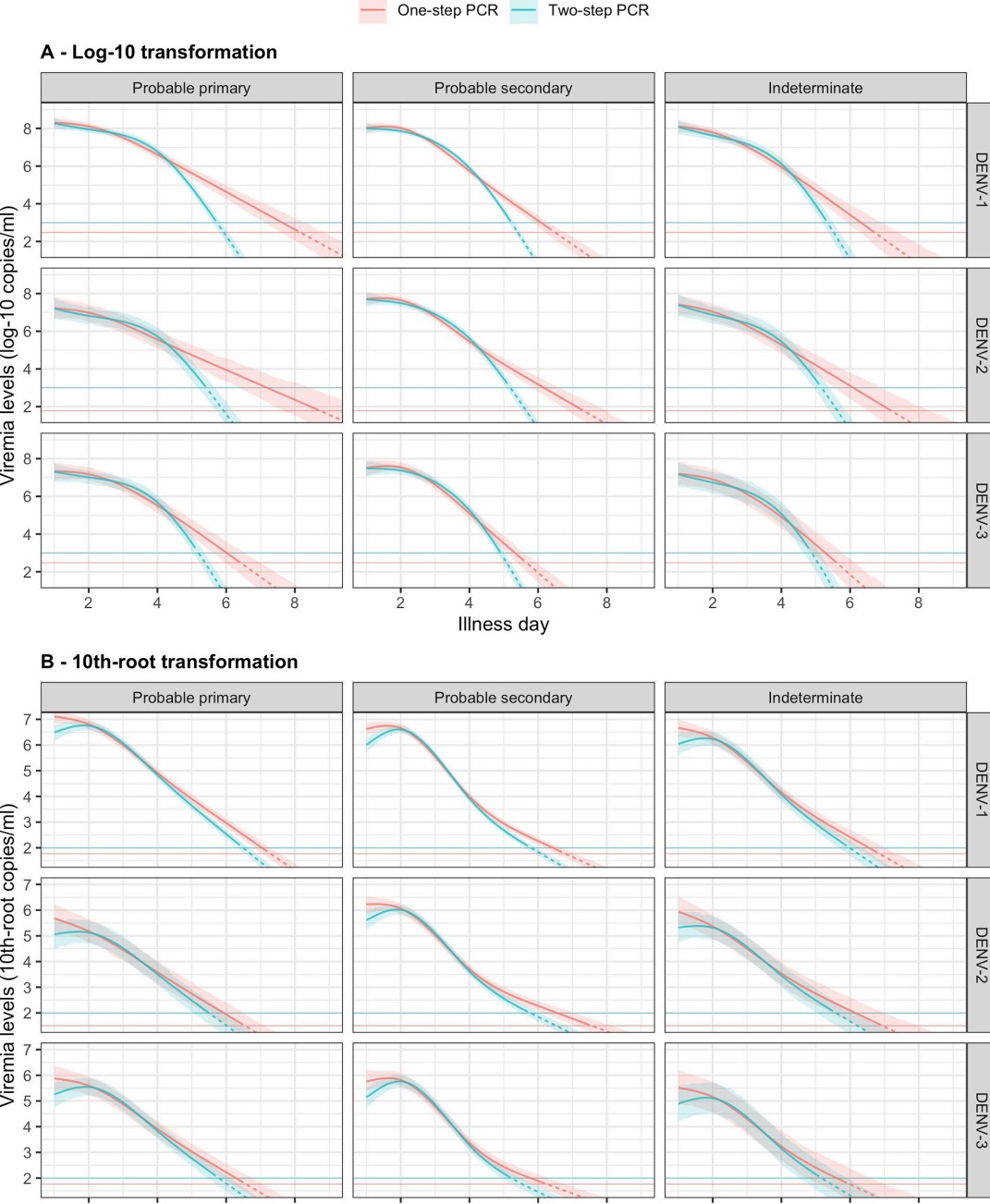

**Appendix 5—figure 1.** Compare fitted trends in mean viremia levels between one- and two-step PCR methods based on (**A**) log-10 and (**B**) 10th-root transformations. The estimated mean viremia levels are represented by colored lines. The 95% credible intervals are shown as colored shaded regions. The horizontal lines indicate the detection limits. Dashed lines indicate the estimated viremia levels below the detection limits. Viremia levels are shown for age of 10 years and male sex. DENV, dengue virus; PCR, polymerase chain reaction.

**Appendix 5—table 1.** p values of parameters in the models of viremia.

| Parameter | p value |
| --- | --- |
| Age (overall effect) | <0.0001 |
| - All interactions of Age | 0.0015 |
| - Nonlinear effect of Age | <0.0001 |
| Sex (overall effect) | 0.0008 |
| - All interactions of Sex | 0.0011 |
| Serotype (overall effect) | <0.0001 |
| - All interactions of Serotype | <0.0001 |
| Immune status (overall effect) | <0.0001 |
| - All interactions of Immune status | <0.0001 |
| PCR method (overall effect) | <0.0001 |
| - All interactions of PCR method | <0.0001 |
| Day (overall effect) | <0.0001 |
| - All interactions of Day | <0.0001 |
| - Nonlinear effect of Day | <0.0001 |
| Serotype * Immune status | 0.0066 |
| Age * Day | 0.0015 |
| Sex * Day | 0.0011 |
| Serotype * Day | <0.0001 |
| Immune status * Day | <0.0001 |
| PCR * Day | <0.0001 |

p values are calculated using the Wald test from the R package 'aod'. This test computes the Wald *chi-squared* test for 1 or more coefficients given their variance–covariance matrix from the model.

A * B is the interaction between A and B.

PCR: polymerase chain reaction.

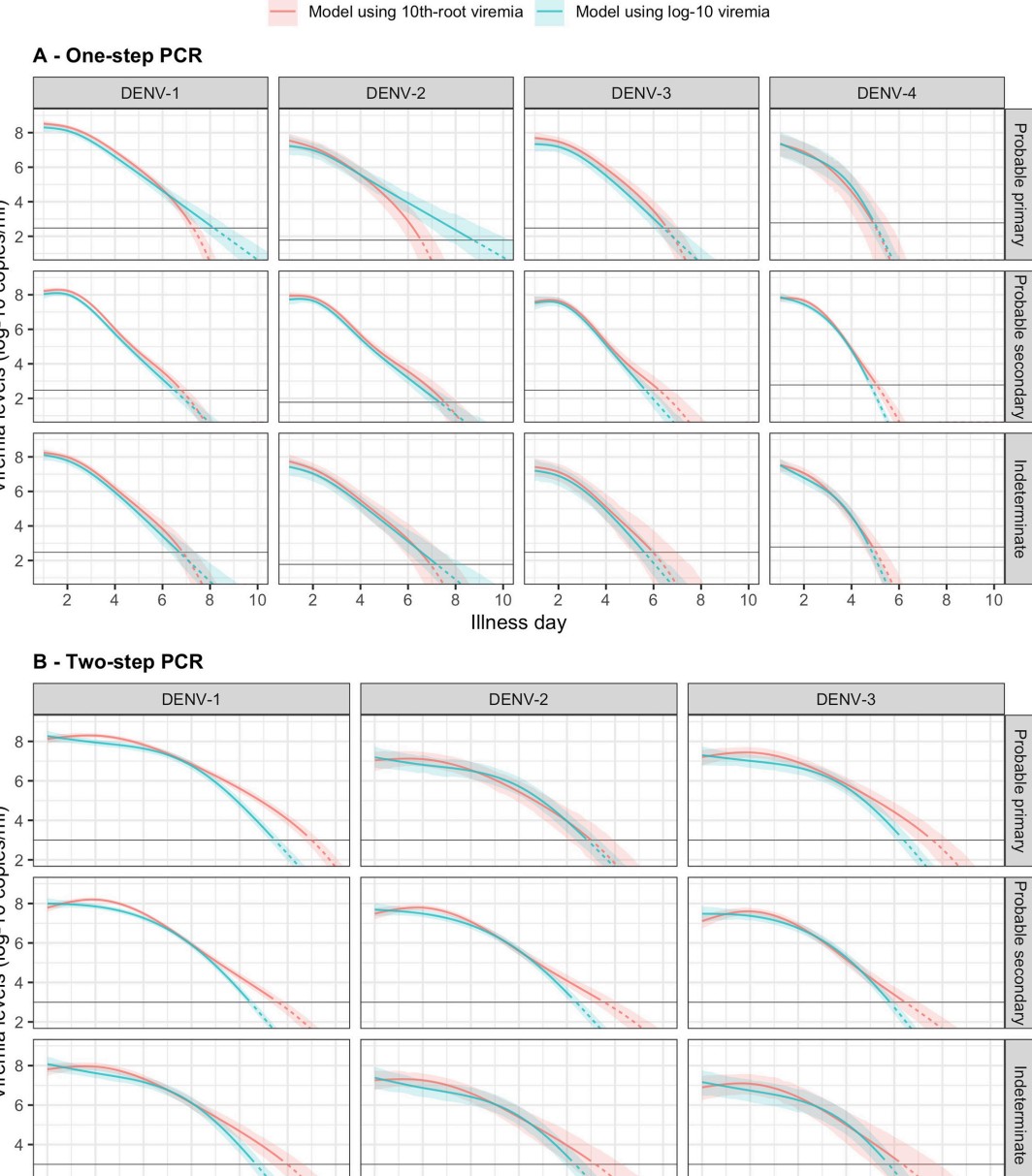

**Appendix 5—figure 2.** Fitted trends in mean viremia levels by models based on 10th-root and log-10 viremia transformations for (**A**) one-step and (**B**) two-step PCR methods. The estimated mean viremia levels are represented by colored lines. The 95% confidence intervals are shown as colored shaded regions. The horizontal lines indicate the detection limits. Dashed lines indicate the estimated viremia levels below the detection limits. Predicted mean viremia levels from 10th-root transformation are transformed to the log-10 scale. The viremia levels are shown for an age of 10 years and male sex.

# Appendix 6

## Results for the effect of viremia on platelet count

**Appendix 6—table 1.** p values of parameters in the supermodel for platelet count.

| Parameter | p value | Parameter | p value |
|---|---|---|---|
| Age (overall effect) | <0.0001 | Age * Day | <0.0001 |
| - All interactions of Age | <0.0001 | Sex * Day | <0.0001 |
| - Nonlinear effect of Age | <0.0001 | Serotype * Day | <0.0001 |
| Sex (overall effect) | <0.0001 | Immune status * Day | <0.0001 |
| - All interactions of Sex | <0.0001 | PCR * Day | <0.0001 |
| Serotype (overall effect) | <0.0001 | Landmark * Day | <0.0001 |
| - All interactions of Serotype | <0.0001 | Viremia * Day | <0.0001 |
| Immune status (overall effect) | <0.0001 | Viremia * Serotype | <0.0001 |
| - All interactions of Immune status | <0.0001 | Viremia * Immune status | 0.0017 |
| PCR method (overall effect) | <0.0001 | Viremia * PCR | <0.0001 |
| - All interactions of PCR method | <0.0001 | Viremia * Landmark | <0.0001 |
| Day (overall effect) | <0.0001 | Serotype * Landmark | 0.0403 |
| - All interactions of Day | <0.0001 | PCR * Landmark | <0.0001 |
| - Nonlinear effect of Day | <0.0001 | Serotype * PCR | <0.0001 |
| Landmark (overall effect) | <0.0001 | Serotype * Immune status | <0.0001 |
| - All interactions of Landmark | <0.0001 | Viremia * Serotype * PCR | 0.2465 |
| - Nonlinear effect of Landmark | <0.0001 | Viremia * Serotype * Day | <0.0001 |
| Viremia (overall effect) | <0.0001 | Viremia * PCR * Day | <0.0001 |
| - All interactions of Viremia | <0.0001 | Viremia * Serotype * Landmark | 0.2601 |
| - Nonlinear effect of Viremia | <0.0001 | Viremia * PCR * Landmark | 0.0846 |
| - Undetectable Viremia | 0.1632 | Viremia * Serotype * Immune status | 0.0589 |
| | | Serotype * PCR * Day | <0.0001 |
| | | Serotype * PCR * Landmark | 0.1347 |
| | | Viremia * Serotype * PCR * Day | 0.4683 |
| | | Viremia * Serotype * PCR * Landmark | 0.3646 |

p values are calculated using the Wald test from the R package 'aod'. This test computes the Wald *chi-squared* test for 1 or more coefficients given their variance–covariance matrix from the model. Note that the overall effect of viremia includes the binary variable describing detectable or undetectable viremia, and the term 'Viremia' in all interactions also consists of that binary variable.

A * B is the interaction between A and B.

PCR: polymerase chain reaction.

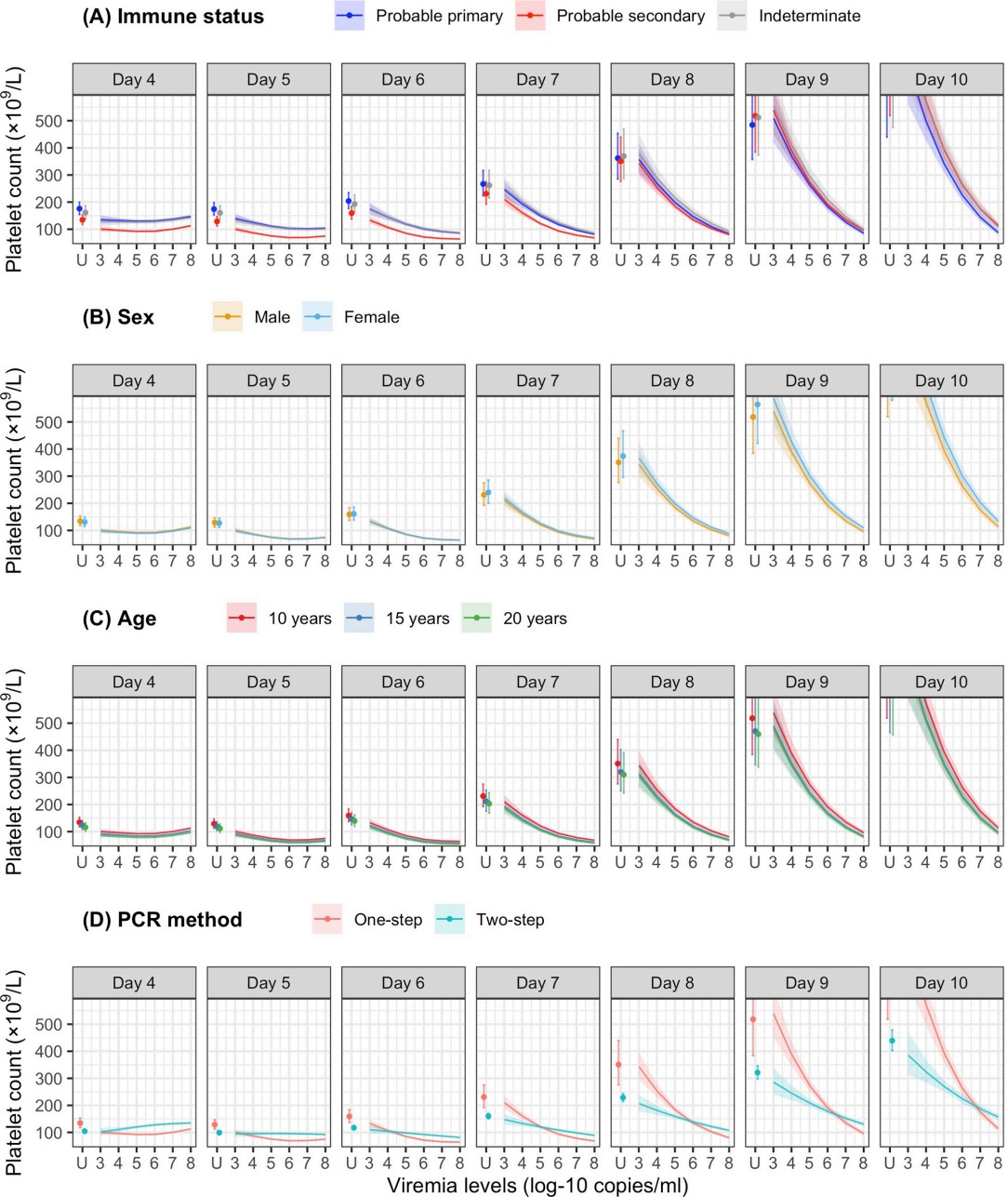

**Appendix 6—figure 1.** Fitted trends in mean platelet counts according to viremia levels by subgroups of the covariates (**A**) immune status, (**B**) sex, (**C**) age, and (**D**) PCR method – results from the supermodel. The colored lines or dots represent the estimated mean platelet counts. The colored shaded regions and whiskers indicate the corresponding 95% confidence intervals. The figures show the effect of viremia on day 4 to platelet count from days 4 to 10. The mean platelet counts are shown for age of 10 years, male sex, serotype DENV-1, probable secondary infection, and using the one-step PCR. DENV, dengue virus; LM, landmark; PCR, polymerase chain reaction; U, under the limit of detection.

# Appendix 7

## Results for the effect of viremia level on clinical outcomes

**Appendix 7—table 1.** Estimates from models of viremia and clinical outcomes.

| | Severe dengue | | Plasma leakage | |
|---|---|---|---|---|
| | OR (95% CI) | p | OR (95% CI) | p |
| Viremia level (log-10 copies/ml)* | | <0.001 | | <0.001 |
| - 7 vs 6 (on day 1) | 2.98 (1.46; 6.10) | | 1.94 (1.50; 2.51) | |
| - 8 vs 7 (on day 1) | 2.27 (1.20; 4.31) | | 1.83 (1.44; 2.32) | |
| - 7 vs 6 (on day 2) | 2.10 (1.32; 3.36) | | 1.62 (1.36; 1.94) | |
| - 8 vs 7 (on day 2) | 1.60 (1.05; 2.44) | | 1.53 (1.29; 1.82) | |
| - 7 vs 6 (on day 3) | 1.51 (1.13; 2.00) | | 1.38 (1.20; 1.57) | |
| - 8 vs 7 (on day 3) | 1.15 (0.84; 1.57) | | 1.30 (1.11; 1.52) | |
| - 7 vs 6 (on day 4) | 1.18 (0.91; 1.54) | | 1.26 (1.10; 1.45) | |
| - 8 vs 7 (on day 4) | 0.90 (0.61; 1.32) | | 1.19 (0.98; 1.44) | |
| - 7 vs 6 (on day 5) | 1.08 (0.75; 1.54) | | 1.31 (1.10; 1.57) | |
| - 8 vs 7 (on day 5) | 0.82 (0.49; 1.37) | | 1.24 (0.96; 1.59) | |
| Undetectable viremia (Yes vs 3 log-10 copies/ml) | 0.48 (0.20; 1.16) | 0.102 | 1.84 (0.51; 6.65) | 0.247 |
| Age (years)* | | 0.182 | | 0.105 |
| - 15 vs 10 | 0.77 (0.50; 1.18) | | 1.07 (0.88; 1.31) | |
| - 20 vs 15 | 0.67 (0.44; 1.04) | | 0.95 (0.85; 1.06) | |
| Sex (Male vs Female) | 1.43 (0.79; 2.61) | 0.011 | 1.10 (0.84; 1.45) | 0.711 |
| Serotype | | <0.001 | | <0.001 |
| - DENV-1 | Ref | | Ref | |
| - DENV-2 | 0.95 (0.44; 2.07) | | 1.22 (0.84; 1.77) | |
| - DENV-3 | 0.35 (0.08; 1.51) | | 0.92 (0.54; 1.56) | |
| - DENV-4 | 2.41 (0.83; 6.98) | | 1.19 (0.76; 1.86) | |
| Immune status | | <0.001 | | <0.001 |
| - Probable primary | 0.02 (0.00; 0.15) | | 0.41 (0.27; 0.63) | |
| - Probable secondary | Ref | | Ref | |
| - Indeterminate | 0.32 (0.04; 2.32) | | 0.57 (0.30; 1.11) | |
| Study | | 0.010 | | <0.001 |
| - Study C | Ref | | Ref | |
| - Study A | 1.36 (0.43; 4.28) | | 1.24 (0.80; 1.93) | |
| - Study B | 2.86 (1.21; 6.80) | | 2.68 (1.92; 3.75) | |
| Landmark (illness day)* | | <0.001 | | <0.001 |
| - 5 vs 3 | 1.38 (0.74; 2.57) | | 1.37 (1.06; 1.78) | |
| - 7 vs 5 | 0.58 (0.17; 2.05) | | 1.34 (0.77; 2.35) | |
| Interaction of viremia† | | | | |
| - With age | | 0.237 | | 0.044 |
| - With sex | | 0.004 | | 0.590 |

*Appendix 7—table 1 Continued on next page*

*Appendix 7—table 1 Continued*

| | Severe dengue | | Plasma leakage | |
|---|---|---|---|---|
| | OR (95% CI) | p | OR (95% CI) | p |
| - With landmark (illness day) | | <0.001 | | <0.001 |
| - With study | | 0.030 | | 0.261 |
| - With serotype | | <0.001 | | <0.001 |
| - With immune status | | <0.001 | | 0.001 |
| - With serotype * immune status | | <0.001 | | 0.012 |
| - With serotype * landmark | | <0.001 | | <0.001 |
| - With immune status * landmark | | <0.001 | | 0.014 |
| - With serotype * immune status * landmark | | <0.001 | | 0.005 |
| Interaction of serotype and immune status | | <0.001 | | <0.001 |
| Nonlinear trend | | | | |
| - Viremia level | | 0.031 | | 0.272 |
| - Age | | 0.349 | | 0.017 |
| - Landmark (illness day) | | <0.001 | | <0.001 |

[*] The model incorporates nonlinear effects of log-10 viremia level, age, and illness day on the endpoints. To facilitate the interpretation of the results, ORs and their corresponding 95% CIs are provided for two selected contrasts of viremia level, age, and landmark illness day.

[†] In the model for plasma leakage, the interactions of viremia include the binary variable describing detectable or undetectable viremia.

Given the complexity of the models involving multiple interactions, ORs and 95% CIs are estimated for specific values of the interacting variables: log-10 viremia of 7, age of 10years, male sex, landmark illness day of 3, serotype DENV-1, and probable secondary infection. CI, confidence interval; DENV, dengue virus; OR, odds ratio; Ref, reference.

| Landmark | N | Severe dengue | Percentage |
|---|---|---|---|
| 1 | 309 | 6 | 1.9 |
| 2 | 1148 | 21 | 1.8 |
| 3 | 2243 | 59 | 2.6 |
| 4 | 2205 | 43 | 2.0 |
| 5 | 2111 | 37 | 1.8 |
| 6 | 1821 | 18 | 1.0 |
| 7 | 1145 | 3 | 0.3 |

| Landmark | N | Plasma leakage | Percentage |
|---|---|---|---|
| 1 | 309 | 37 | 12.0 |
| 2 | 1148 | 178 | 15.5 |
| 3 | 2239 | 343 | 15.3 |
| 4 | 2155 | 319 | 14.8 |
| 5 | 1998 | 247 | 12.4 |
| 6 | 1619 | 135 | 8.3 |
| 7 | 937 | 39 | 4.2 |

Detectable viremia for severe dengue

Detectable viremia for plasma leakage

Undetectable viremia for severe dengue

Undetectable viremia for plasma leakage

**Appendix 7—figure 1.** Frequency of the outcomes and regression coefficients for individuals observed on each landmark day that did not have the outcome yet. Relationship between viremia and outcome. A logistic regression model with covariates viremia level (with linear trend) and the binary variable describing whether the viremia value was undetectable is fitted per landmark time point. Solid lines are the regression coefficients and dashed lines are the 95% confidence intervals.

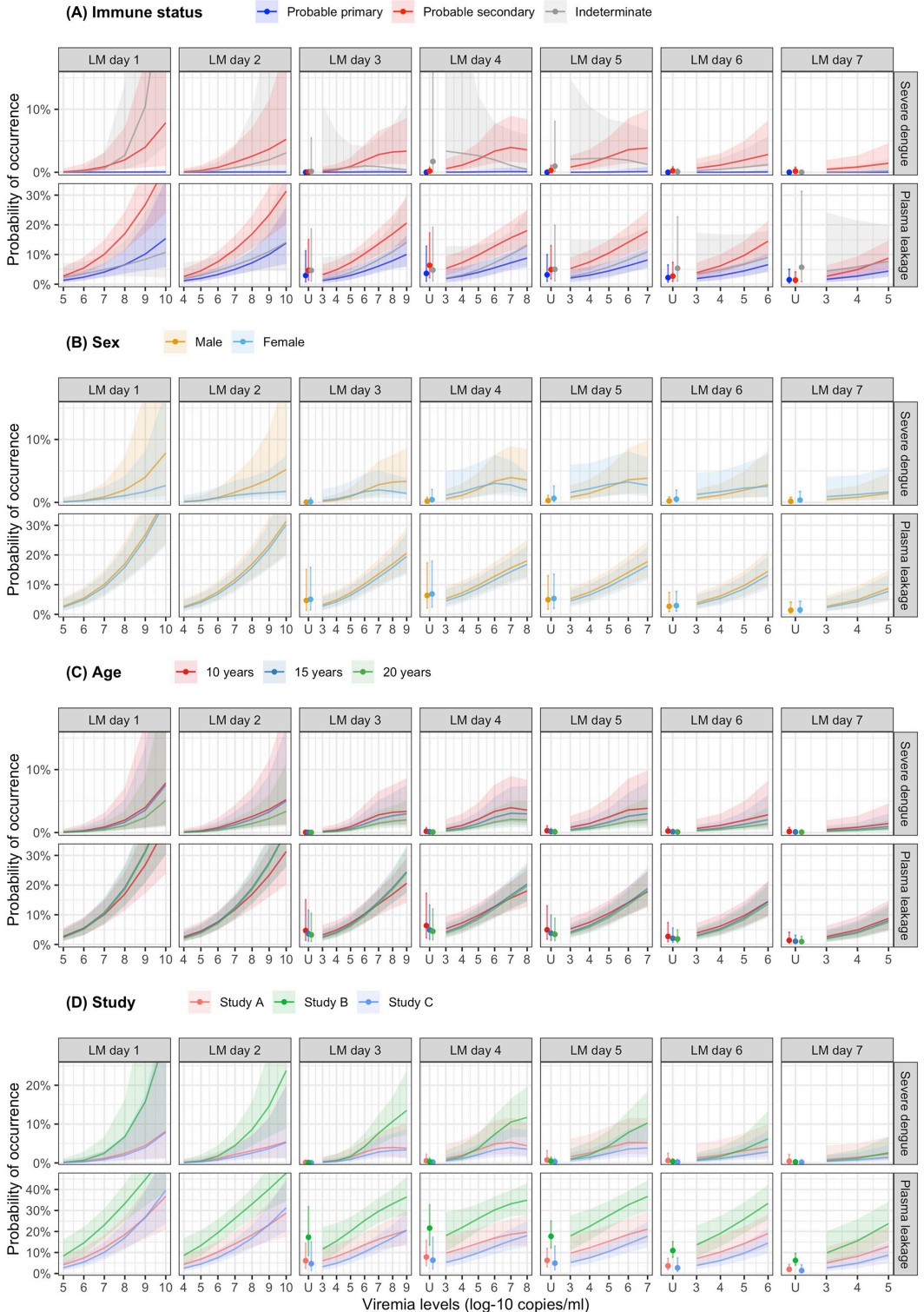

**Appendix 7—figure 2.** Probability of occurrence of the two clinical endpoints according to viremia levels by subgroups of the covariates (**A**) immune status, (**B**) sex, (**C**) age, and (**D**) study – results from the supermodel. The colored lines or dots represent the probability of the endpoints. The colored shaded regions and whiskers indicate the corresponding 95% confidence intervals. Each column represents the effect of viremia on a specific day. Note that the probability of severe dengue in the probable primary group is almost 0% because there was only one case with severe dengue in this group. The probabilities are shown for age of 10 years, male sex, serotype DENV-1, probable secondary infection, and from study C. LM, landmark; U, under the limit of detection.

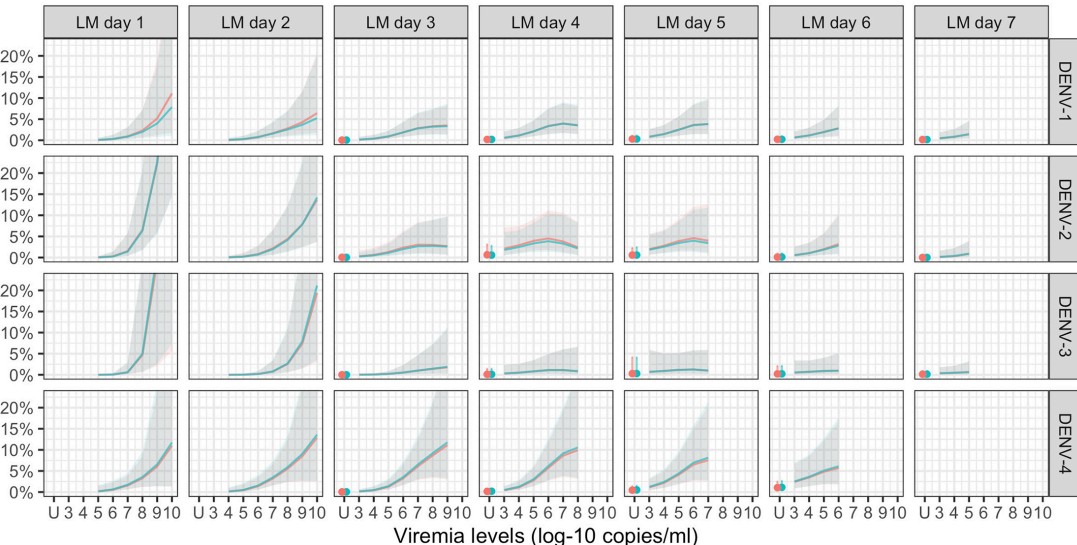

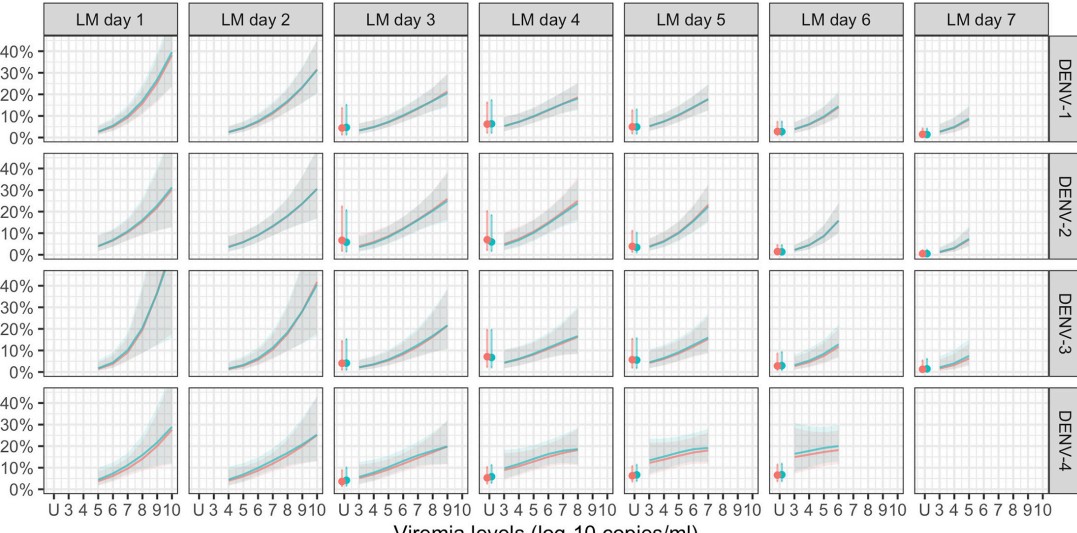

**Appendix 7—figure 3.** Probability of occurrence of (**A**) severe dengue and (**B**) plasma leakage according to viremia levels – compare results from the supermodels with and without multiple imputation. The colored lines or dots represent the probability of the endpoints. The colored shaded regions and whiskers indicate the corresponding 95% confidence intervals. Each column represents the effect of viremia on a specific day. No fitted trends are made for DENV-4 in LM day 7 since viremia was undetectable in almost all DENV-4 cases from day 7 onwards. The probabilities are shown for age of 10 years, male sex, probable secondary infection, and from study C. DENV, dengue virus; LM, landmark; PCR, polymerase chain reaction; U, under the limit of detection.

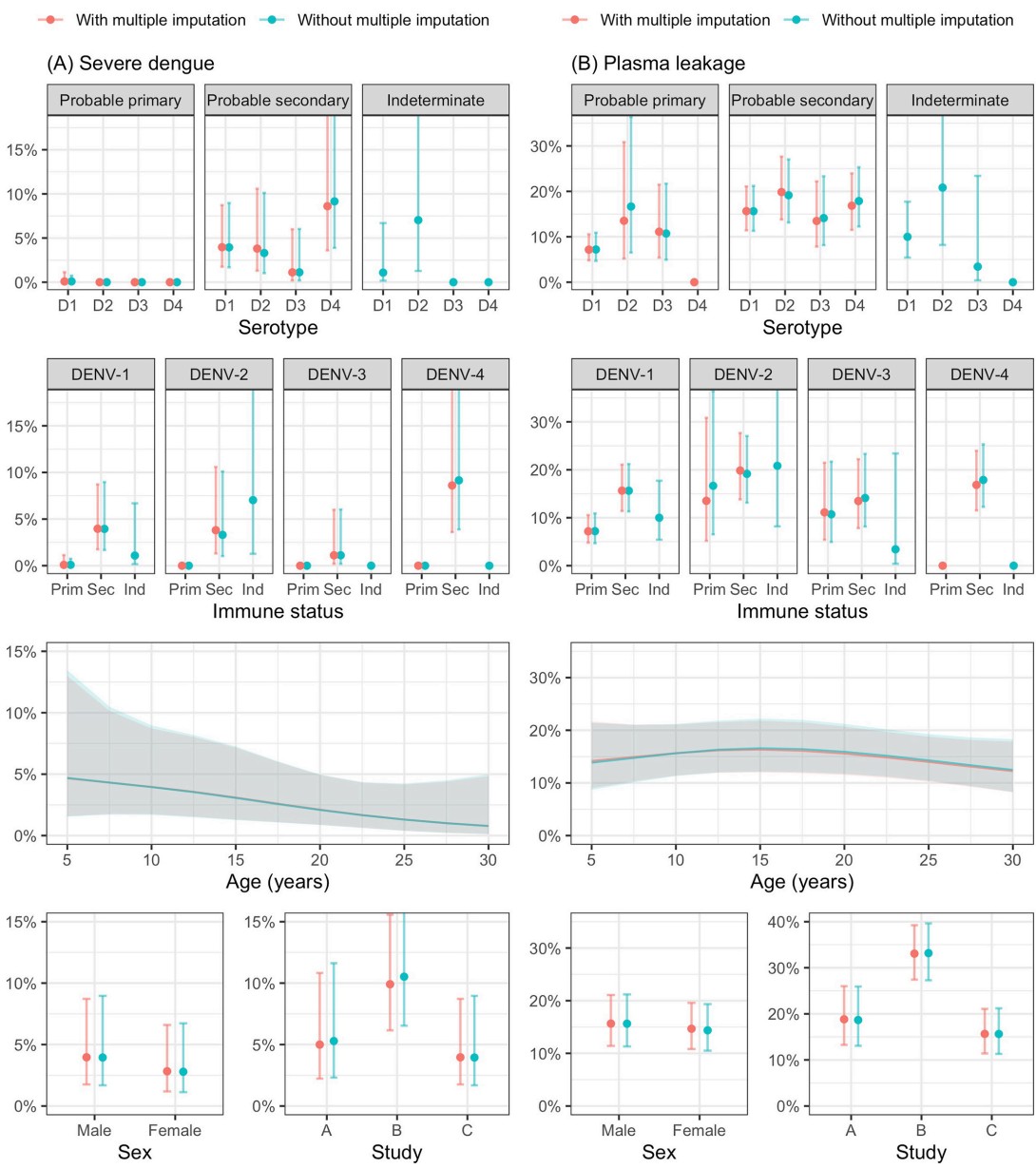

**Appendix 7—figure 4.** Probability of occurrence of (**A**) severe dengue and (**B**) plasma leakage according to the covariates – results from the supermodel. The lines and dots represent the probability of the endpoints. The shaded regions and whiskers indicate the corresponding 95% confidence intervals. Note that the probability of severe dengue in the probable primary group is almost 0% because there was only one case with severe dengue in this group. The probabilities are shown for age of 10 years, male sex, serotype DENV-1, probable secondary infection, viremia level of 7 log-10 copies/ml, landmark illness day 4, and from study C. D1, D2, D3, and D4 are DENV-1, DENV-2, DENV-3, and DENV-4, respectively; Prim, probable primary infection; Sec, probable secondary infection; Ind., indeterminate immune status.

**Appendix 7—table 2.** Estimates from models of the rate of decline in viremia and clinical outcomes.

| | Severe dengue | | Plasma leakage | |
|---|---|---|---|---|
| | **OR** (95% CI) | **p** | **OR** (95% CI) | **p** |
| Rate of decline in viremia (per 0.5 log-10 copies/ml increase) | 0.14 (0.06; 0.33) | <0.001 | | <0.001 |
| - DENV-1 – probable primary | | | 0.23 (0.09; 0.56) | |

*Appendix 7—table 2 Continued on next page*

*Appendix 7—table 2 Continued*

| | Severe dengue | | Plasma leakage | |
|---|---|---|---|---|
| | **OR** (95% CI) | **p** | **OR (95% CI)** | **p** |
| - DENV-1 – probable secondary | | | 0.12 (0.05; 0.28) | |
| - DENV-2 – probable primary | | | 0.30 (0.11; 0.80) | |
| - DENV-2 – probable secondary | | | 0.16 (0.06; 0.38) | |
| - DENV-3 – probable primary | | | 0.18 (0.06; 0.54) | |
| - DENV-3 – probable secondary | | | 0.09 (0.03; 0.27) | |
| - DENV-4 – probable primary | | | 0.78 (0.24; 2.50) | |
| - DENV-4 – probable secondary | | | 0.40 (0.16; 1.04) | |
| Serotype | | 0.001 | | 0.006 |
| - DENV-1 | Ref | | Ref | |
| - DENV-2 | 0.74 (0.39; 1.39) | | 1.04 (0.75; 1.45) | |
| - DENV-3 | 0.21 (0.05; 0.88) | | 0.81 (0.54; 1.22) | |
| - DENV-4 | 8.59 (2.38; 31.03) | | 4.84 (2.07; 11.32) | |
| Immune status | | <0.001 | | <0.001 |
| - Probable primary | Ref | | Ref | |
| - Probable secondary | 63.23 (8.15; 490.31) | | 2.77 (1.78; 4.33) | |
| - Indeterminate | 16.29 (1.77; 150.25) | | 1.39 (0.68; 2.82) | |
| PCR method | | <0.001 | | <0.001 |
| - One-step | Ref | | Ref | |
| - Two-step | 33.32 (10.92; 101.66) | | 14.62 (6.98; 30.60) | |
| Interaction of rate of decline in viremia | | | | |
| - With serotype | | | | 0.182 |
| - With immune status | | | | 0.017 |
| - With PCR | | | | 0.006 |

The model for severe dengue does not include any interaction term.

In the model of plasma leakage, interactions between the rate of decline in viremia and serotype, immune status, and PCR are included. ORs and 95% CIs of the rate of decline in viremia are estimated for the one-step PCR in each subgroup of serotype and probable primary/secondary infection. ORs and 95% CIs of serotype, immune status, and PCR are estimated for the rate of decline in viremia of 1.4 log-10 copies/ml.

CI, confidence interval; DENV, dengue virus; OR, odds ratio; PCR, polymerase chain reaction; Ref, reference.

## Appendix 8

### STROBE Statement – checklist of items

Dengue viremia kinetics and relationship with platelet count and clinical outcomes: an analysis of 2340 patients from Vietnam

| | Item No | Recommendation | Comments or page(s) reported |
|---|---|---|---|
| | | (*a*) Indicate the study's design with a commonly used term in the title or the abstract | Page 1 |
| Title and abstract | 1 | (*b*) Provide in the abstract an informative and balanced summary of what was done and what was found | Page 2 |
| **Introduction** | | | |
| Background/rationale | 2 | Explain the scientific background and rationale for the investigation being reported | Pages 3–4 |
| Objectives | 3 | State specific objectives, including any prespecified hypotheses | Page 4 |
| **Methods** | | | |
| Study design | 4 | Present key elements of study design early in the paper | Pages 4–5 |
| Setting | 5 | Describe the setting, locations, and relevant dates, including periods of recruitment, exposure, follow-up, and data collection | Pages 4–5, Appendix 1 |
| | | (*a*) Give the eligibility criteria, and the sources and methods of selection of participants. Describe methods of follow-up | Pages 4–5, Appendix 1 |
| Participants | 6 | (*b*) For matched studies, give matching criteria and number of exposed and unexposed | Not applicable |
| Variables | 7 | Clearly define all outcomes, exposures, predictors, potential confounders, and effect modifiers. Give diagnostic criteria, if applicable | Page 5, Appendix 2 |
| Data sources/ measurement | 8* | For each variable of interest, give sources of data and details of methods of assessment (measurement). Describe comparability of assessment methods if there is more than one group | Page 5 |
| Bias | 9 | Describe any efforts to address potential sources of bias | Pages 5–7 |
| Study size | 10 | Explain how the study size was arrived at | Pages 5–7, *Figure 1* |
| Quantitative variables | 11 | Explain how quantitative variables were handled in the analyses. If applicable, describe which groupings were chosen and why | Pages 6–7, Appendix 3 |
| | | (*a*) Describe all statistical methods, including those used to control for confounding | Pages 6–7, Appendix 3 |
| | | (*b*) Describe any methods used to examine subgroups and interactions | Pages 6–7, Appendix 3 |
| | | (*c*) Explain how missing data were addressed | Page 7, Appendix 3 |
| | | (*d*) If applicable, explain how loss to follow-up was addressed | Not available |
| Statistical methods | 12 | (*e*) Describe any sensitivity analyses | Pages 6–7, Appendix 3 |
| **Results** | | | |
| | | Report numbers of individuals at each stage of study—eg numbers potentially eligible, examined for eligibility, confirmed eligible, included in the study, completing follow-up, and analysed | Pages 5–6, *Figure 1* |
| | | Give reasons for non-participation at each stage | Pages 5–6, *Figure 1* |
| Participants | 13* | Consider use of a flow diagram | *Figure 1* |
| | | Give characteristics of study participants (eg demographic, clinical, social) and information on exposures and potential confounders | Pages 7–8, *Table 1* |
| | | Indicate number of participants with missing data for each variable of interest | Pages 7–8, *Table 1* |
| Descriptive data | 14* | Summarise follow-up time (eg, average and total amount) | Not available |
| Outcome data | 15* | Report numbers of outcome events or summary measures over time | Pages 7–8, *Table 1*, *Appendix 4—table 1* |
| | | (*a*) Give unadjusted estimates and, if applicable, confounder-adjusted estimates and their precision (eg, 95% confidence interval). Make clear which confounders were adjusted for and why they were included | Pages 8–9, Appendices 5–7 |
| | | (*b*) Report category boundaries when continuous variables were categorized | Not available |
| Main results | 16 | (*c*) If relevant, consider translating estimates of relative risk into absolute risk for a meaningful time period | Pages 8–9, Appendices 5–7 |

*Continued on next page*

*Continued*

| | Item No | Recommendation | Comments or page(s) reported |
|---|---|---|---|
| Other analyses | 17 | Report other analyses done—eg analyses of subgroups and interactions, and sensitivity analyses | Pages 8–9, Appendices 5–7 |
| **Discussion** | | | |
| Key results | 18 | Summarise key results with reference to study objectives | Page 10 |
| Limitations | 19 | Discuss limitations of the study, taking into account sources of potential bias or imprecision. Discuss both direction and magnitude of any potential bias | Page 12 |
| Interpretation | 20 | Give a cautious overall interpretation of results considering objectives, limitations, multiplicity of analyses, results from similar studies, and other relevant evidence | Pages 10–12 |
| Generalisability | 21 | Discuss the generalisability (external validity) of the study results | Pages 11–12 |
| **Other information** | | | |
| Funding | 22 | Give the source of funding and the role of the funders for the present study and, if applicable, for the original study on which the present article is based | Page 14 |

*Give information separately for exposed and unexposed groups.

Note: An Explanation and Elaboration article discusses each checklist item and gives methodological background and published examples of transparent reporting. The STROBE checklist is best used in conjunction with this article (freely available on the Web sites of PLoS Medicine at http://www.plosmedicine.org/, Annals of Internal Medicine at http://www.annals.org/, and Epidemiology at http://www.epidem.com/). Information on the STROBE Initiative is available at http://www.strobe-statement.org.

