## [Editor Report · eLife assessment]

This manuscript by Vuong and colleagues reports on the kinetics of viremia in a large set of individuals from Vietnam. In the large cohort, all 4 dengue serotypes are represented and the authors try to correlate viraemia measured at various days from illness onset with thrombocytopaenia and severe dengue, according to the WHO 2009 classification scheme. These are **fundamental** findings that provide **compelling** evidence of the importance of measuring viremia early in the phase of the disease. These data will help to inform the design of studies of antiviral drugs against dengue.

---

## [Referee Report · Reviewer #1 (Public review)]

Summary:

This manuscript by Vuong and colleagues reports a study that pooled data from 3 separate longitudinal study that collectively spanned an observation period of over 15 years. The authors examined for correlation between viraemia measured at various days from illness onset with thrombocytopaenia and severe dengue, according to the WHO 2009 classification scheme. The motivation for this study is both to support the use of viraemia measurement as a prognostic indicator of dengue and also to, when an antiviral drug becomes licensed for use, guide the selection of patients for antiviral therapy. They found that the four DENVs show differences in peak and duration of viraemia and that viraemia levels before day 5 but not those after from illness onset correlated with platelet count and plasma leakage at day 7 onwards. They concluded that the viraemia kinetics call for early measurement of viraemia levels in the early febrile phase of illness.

Strengths:

This is a unique study due to the large sample size and longitudinal viraemia measurements in the study subjects. The data addresses a gap in information in the literature, where although it has been widely indicated that viraemia levels are useful when collected early in the course of illness, this is the first time anyone has systematically examined this notion. The inclusion of correlation between rate of viraemia decline and risk of severe dengue/plasma leakage further strengthens the relevance of this paper to those interested in anti-dengue therapeutic research and development.

Weaknesses:

The study only analysed data from dengue patients in Vietnam. Moreover, the majority of these patients had DENV-1 infection; few had DENV-4 infection. The data could thus be skewed by the imbalance in the prevalence of the different types of DENV during the period of observation. The use of patient-reported time of symptom onset as a reference point for viraemia measurement is pragmatic although there is subjectivity and thus noise in the data.

---

## [Referee Report · Reviewer #2 (Public review)]

Summary:

The authors have carried out a comprehensive analysis regarding the kinetics of viraemia and clinical disease severity.

Strengths:

The manuscript provides important information, especially regarding the time of clearance of the virus and disease severity.

Weaknesses:

Due to the lower number of patients with primary dengue, cannot get an idea regarding viraemia kinetics and disease severity for different serotypes during primary infection.

---

## [Author Response]

The following is the authors’ response to the original reviews.

**Reviewer #1 (Public Review):**
Summary:This manuscript by Vuong and colleagues reports a study that pooled data from 3 separate longitudinal studies that collectively spanned an observation period of over 15 years. The authors examined for correlation between viraemia measured at various days from illness onset with thrombocytopaenia and severe dengue, according to the WHO 2009 classification scheme. The motivation for this study is both to support the use of viraemia measurement as a prognostic indicator of dengue and also when an antiviral drug becomes licensed for use, to guide the selection of patients for antiviral therapy. They found that the four DENVs show differences in peak and duration of viraemia and that viraemia levels before day 5 but not those after from illness onset correlated with platelet count and plasma leakage at day 7 onwards. They concluded that the viraemia kinetics call for early measurement of viraemia levels in the early febrile phase of illness.Strengths:This is a unique study due to the large sample size and longitudinal viraemia measurements in the study subjects. The data addresses a gap in information in the literature, where although it has been widely indicated that viraemia levels are useful when collected early in the course of illness, this is the first time anyone has systematically examined this notion.Weaknesses:The study only analysed data from dengue patients in Vietnam. Moreover, the majority of these patients had DENV-1 infection; few had DENV-4 infection. The data could thus be skewed by the imbalance in the prevalence of the different types of DENV during the period of observation. The use of patient-reported time of symptom onset as a reference point for viraemia measurement is pragmatic although there is subjectivity and thus noise in the data.

We acknowledge and appreciate your comments regarding the limitations of our study, including the pooled data from Vietnam and the use of symptom onset as a reference point for viremia kinetics. These points have been incorporated into the “Limitations” section.

**Reviewer #2 (Public Review):**
Summary:This manuscript highlights very important findings in the field, especially in designing clinical trials for the evaluation of antivirals.Strengths:The study shows significant differences between the kinetics of viral loads between serotypes, which is very interesting and should be taken into account when designing trials for antivirals.Weaknesses:The kinetics of the viral loads based on disease severity throughout the illness are not described, and it would be important if this could be analyzed.

In response to your suggestion, we have expanded our analysis to investigate the relationship between the rate of viremia decline and clinical outcomes. Our findings demonstrate that a faster rate of viremia decline is associated with a reduced risk of severe clinical outcomes. We have incorporated this new analysis into the revised manuscript, providing further details in the “Statistical Analysis” section (page 7) and presenting the results on pages 15 and in Figure 6.

**Reviewer #1 (Recommendations For The Authors):**
Several areas require additional attention. I have limited my comments on the findings as I am not a mathematician and cannot knowledgeably comment on the statistical modelling methods.Comment #1: Lines 83-84. Although viraemia level shows declining trends from illness onset and thus lessens its prognostic value, it remains unknown if a more rapid rate of decline in viraemia is associated with a reduced risk of severe dengue. This is the fundamental premise of antiviral drug development for the treatment of dengue. The authors are uniquely poised to show if this logic that underpins antiviral development is likely correct and perhaps even estimate the extent to which a decline in viraemia needs to occur for a measurable reduction in the risk of severe dengue. Could the authors consider such an analysis?

We appreciate your valuable suggestion. In response, we have expanded our analysis to investigate the relationship between the rate of viremia decline and clinical outcomes Utilizing a model of viremia kinetics with the assumption of a linear log-10 viremia decrease over time, we calculated the rate of decline for each patient. Our findings demonstrate that a faster rate of viremia decline is associated with a significantly reduced risk of severe clinical outcomes. We have incorporated this new analysis into the revised manuscript, providing further details in the “Statistical Analysis” section (page 7) and presenting the results on pages 15 and in Figure 6.

Comment #2: Lines 101-102. Studies A and B were conducted in parallel, and several patients enrolled in study A from primary healthcare clinics were eventually also enrolled in study B upon hospitalization. It would be helpful to know how many patients from study A were included in study B. It would also be useful for the authors to indicate if such inclusion would constitute double-counting at any point in their analyses.

To address potential confusion regarding patient overlap between studies A and B, we have provided further clarification in the revised manuscript’s Legend of Figure 1. Among confirmed dengue patients, 31 individuals enrolled in study A were later included in study B upon hospitalization. Of these, 9 had viremia measurements available in both studies and were consequently analysed in study A only. The remaining 22 lacked viremia data in study A but had measurements in study B, leading to their inclusion in study B in the analysis. We have taken meticulous care to ensure no patient data is double-counted.

Comment #3: Lines 126-127. The definition of probable primary and secondary dengue from IgG measurements needs more detail. How was the anti-DENV IgG ELISA data from paired sera interpreted?

To ensure clarity, we have moved the definitions of probable primary and secondary infections from the supplementary file (Appendix 2) to the main text of the revised manuscript (Methods section – Plasma viremia measurement, dengue diagnostics, and clinical endpoints – page 6): “A probable primary infection was defined by two negative/equivocal IgG results on separate samples taken at least two days apart within the first ten days of symptom onset, with at least one sample during the convalescent phase (days 6-10). A probable secondary infection was defined by at least one positive IgG result during the first ten days. Cases without time-appropriate IgG results were classified as indeterminate.”

Comment #4: Lines 230-232 and Figure 4. The findings reported in Figure 4 are curious. Why is the platelet count highest (significantly?) for DENV-1 compared to other DENV-type infections at low viraemia levels on LM days 1-3? Does that also mean that DENV-3 and -4 infections have a greater impact on platelet counts at days 7-10 than DENV-1 and -2?

In our analyses, we allowed the relation between viremia and platelet count to differ by serotype. Figure 4 shows the highest platelet counts for DENV-1 compared to other serotypes, especially at low viremia levels. Apparently, while DENV-1 on average has higher viremia (Figure 3), the same viremia level in DENV-1 compared to other serotypes is associated with a less severe disease course and higher platelet count. This does not necessarily imply that platelet count overall, uncorrected for viremia level, differs by genotype. Indeed, our unpublished analysis (shown below) indicates a modest influence of serotype on platelet count.

Comment #5: Figure 5. In a recent paper (Vuong et al, Clin Infect Dis 2021), the authors show elegantly that the viraemia levels on admission correlated with severe dengue. However, these correlations were different for each of the four DENV types and whether the infection was primary or secondary. Why wasn't the analysis in Figure 5 further stratified by their probable primary or secondary dengue status?

We appreciate your feedback and have stratified Figure 5 by serotype and immune status as suggested. Please note that due to the limited number of severe dengue in primary infections (only 1 case in DENV-1) and plasma leakage in primary DENV-4 (see Appendix 4-table 1), the estimated probability of having these outcomes is nearly zero across all viremia levels within these subgroups.

Comment #6: Line 279. The description in this line is at odds with the data in Figure 3A, which shows that DENV-2 could be detected over a longer period than DENV-1 as the one-step RT-qPCR assay has a lower detection limit than DENV-1.

In response to your feedback, we have revised the description to clarify that DENV-1 exhibits higher viremia levels compared to DENV-2 and DENV-3 in the revised manuscript (page 18).

**Reviewer #2 (Recommendations For The Authors):**
IntroductionComment #1: Line 56: the authors state that viraemia is associated with dengue disease severity and cite their previous results. They then summarize the results of this study and others. The highlights of this paper should be described in more detail. It is important that the authors state the conclusions of their own paper, including that the association was not very strong and that the viral loads were lowest with DENV2, but DENV2 was associated with more severe disease.

Thank you for your comment. To improve the introduction’s flow, we have removed that sentence in line 56 of the manuscript and have added the weak association in the next paragraph (pages 3-4).

Comment #2: It would be important to cite smaller studies that show a delay in clearance of the virus being associated with more severe disease outcomes.

Thanks for your suggestion. We have added information to the introduction (page 4), highlighting a study which found a slower rate of viral clearance to be associated with more severe outcomes (Wang et al., 2008). However, other studies have shown no association (Vaughn et al., 2000; Fox et al., 2011). This lack of conclusive evidence underscores the need for further research.

MethodsComment #3: The authors highlight the possible discrepancies in comparing viral kinetics of two RT-PCR methods. Although it is not ideal to combine such results, the authors have analyzed them separately, providing valuable data.

We appreciate your comment.

Comment #4: Which tests were used to define the immune status as primary and secondary? What were the definitions?

We have moved the definitions of probable primary and secondary infections from the supplementary file (Appendix 2) to the main text of the revised manuscript (Methods section – Plasma viremia measurement, dengue diagnostics, and clinical endpoints – page 6): “A probable primary infection was defined by two negative/equivocal IgG results on separate samples taken at least two days apart within the first ten days of symptom onset, with at least one sample during the convalescent phase (days 6-10). A probable secondary infection was defined by at least one positive IgG result during the first ten days. Cases without time-appropriate IgG results were classified as indeterminate.”

ResultsComment #5: It is interesting that DENV2 showed the slowest decline, but yet associated with overall lower viral loads during early illness and more severe disease outcomes. Could delayed clearance of the virus be associated with disease severity?

We have expanded our analysis to investigate the relationship between the rate of viremia decline and clinical outcomes Utilizing a model of viremia kinetics with the assumption of a linear log-10 viremia decrease over time, we calculated the rate of decline for each patient. Our findings demonstrate that a faster rate of viremia decline is associated with a significantly reduced risk of severe clinical outcomes. We have incorporated this new analysis into the revised manuscript, providing further details in the “Statistical Analysis” section (page 7) and presenting the results on pages 15 and in Figure 6.

Comment #6: Were there any differences in the kinetics of viral loads in children vs adults? I.e. children, young adults and older adults (>60 or 50?). Or were there insufficient numbers for this comparison?

To address this point, we have modified the reported results of Figure 3-D by ages of 5, 10, 15, 25, and 50 years, represented children, adolescents, young adults, and older adults. Our analysis shows that viremia kinetics are largely similar across ages.

Comment #7: Did any patients have comorbidities such as diabetes, obesity etc... if so, were there any differences in the viral loads?

We appreciate your interest in the potential impact of comorbidities on viral loads. However, due to data limitations, we were unable to analyze this association. Only 6 patients had documented diabetes in the pooled dataset. In study C, 39 patients had obesity, whereas body mass index data is not available for studies A and B, although reports suggest a lower prevalence of obesity compared to study C.

Comment #8: Were there any differences in the kinetics of the overall viral loads between DF/DHF/DSS or dengue with warning signs, without warning signs and severe dengue? Especially related to the time for viral clearance?

Thank you for your suggestion. Such analysis reverses time and the causal direction, while we are more interested in looking forward. Therefore, instead of analyzing viremia kinetics based on disease severity, we have added an analysis to investigate the relationship between the rate of decline in viremia and clinical outcomes, as shown in the response to your comment #5. Results show that a more rapid rate of viremia decline is associated with a reduced risk of more severe clinical outcomes. In addition, in this study, we selected two clinical outcomes severe dengue and plasma leakage. The definitions are based on the WHO 2009 guidelines and standard endpoint definitions for dengue trials (Tomashek et al., 2018).